# PREDICTIVE CODING BEYOND CORRELATIONS

## ABSTRACT

Bayesian and causal inference are fundamental processes for intelligence. Bayesian inference describes *observations*: what can be inferred about $y$ if we observe a related variable $x$? Causal inference models *interventions*: if we directly change $x$, how will $y$ change? Predictive coding is a neuroscience-inspired method for performing Bayesian inference on continuous state variables using local information only. In this work, we show how a simple change in the inference process of predictive coding enables interventional and counterfactual inference in scenarios where the causal graph is known. We then extend our results, and show how predictive coding can be used in cases where the graph is unknown, and has to be inferred from observational data. This allows us to perform structure learning and causal query answering on predictive coding-based structural causal models. Empirically, we test our method on a large number of benchmarks, as well as presenting experiments that show potential applications in machine learning.

## 1 INTRODUCTION

Predictive coding (PC) is an influential theory of learning and perception in the brain (Rao & Ballard, 1999; Salvatori et al., 2023; Millidge et al., 2021), with roots in Bayesian inference and signal compression. Conventional literature primarily deals with hierarchical models relating top-down predictions to internal states and external stimuli (Rao & Ballard, 1999; Friston, 2005; 2008; Whittington & Bogacz, 2017). A recent work went beyond this, and has shown how PC can be used to perform inference and learning on structures with any topology (Salvatori et al., 2022a). A natural consequence is that PC can be used to perform Bayesian inference on models with more entangled structures, such as directed graphical models with cyclic structures. This kind of inference is, however, limited to the computation of conditional probabilities, i.e., correlations. In this work, we question whether PC models have the capabilities to go beyond the computation of correlations, and show how it is possible to use the same model for causal inference (Pearl, 2009).

Research on causality is divided into two primary areas: causal inference, which aims to infer the effect of an intervention in a known system, and causal discovery, which aims to discover the causal graph underlying observational data. Here, we tackle both tasks, by first showing how PC is able to naturally model interventions using a differentiable framework that aims to minimize a variational free energy (Friston, 2005; Rao & Ballard, 1999), with only a simple adjustment to their standard Bayesian inference procedure, and then showing how to use the same framework to perform structure learning from observational data (up to an equivalence class, called Markov equivalence class). This shows that PC is an end-to-end causality engine, able to answer causal queries without previous knowledge of the parent-child relationships.The main goal of our work is not to solve open problems in the causality literature, but to show how it is possible to model interventions in a biologically plausible and efficient fashion, without the need of mutilating a graph, as it is instead done in Fig. 1.

The rest of this work has the following structure: in Section 2, we review the basic concepts of Bayesian networks, and their connection with Pearlian's causality. Then, we show how the PC framework developed to train graphs with arbitrary topologies (Salvatori et al., 2022a) can be used to perform conditional inference on Bayesian networks. In Section 3, we show how the same model can compute interventions by setting the prediction error of a specific node to zero during the inference process. Empirically, we test our claims on PC-based structural causal models, and show promising results on machine learning and causal query benchmarks (De Brouwer, 2022). In Section 4, we show how PC graphs can perform structure learning from observational data.

## 2 BAYESIAN NETWORKS AND PREDICTIVE CODING

Assume we have a set of $N$ random variables $\mathbf{X} = \{\mathbf{x}_1, \ldots, \mathbf{x}_N\}$, with $\mathbf{x}_i \in \mathbb{R}^d$. Relations among variables are represented by a directed graph $G = (V, E)$, also called *causal graph* of $S$. Every vertex $v_i \in V$ represents a random variable $\mathbf{x}_i$, and every edge $(i, j)$ represents a causal relation from $\mathbf{x}_i$ to $\mathbf{x}_j$. The causal graph defines the joint distribution of the system, computed as follows:

Figure 1: Example socio-economic graph and its structure after conditioning and intervening on *education level*.

$$p(\mathbf{x}_1, \ldots, \mathbf{x}_N) = \Pi_{i=1}^N \, p(\mathbf{x}_i | par(\mathbf{x}_i)),$$

with $par(\mathbf{x}_i)$ being the parent nodes of $\mathbf{x}_i$. In Fig. 1, we show a graph with joint probability

$$p(\mathbf{x}_1, \mathbf{x}_2, \mathbf{x}_3, \mathbf{x}_4) = p(\mathbf{x}_1)p(\mathbf{x}_2)p(\mathbf{x}_3 \mid \mathbf{x}_1, \mathbf{x}_2)p(\mathbf{x}_4 \mid \mathbf{x}_2, \mathbf{x}_3).$$

Given the causal graph on the left, the arrows indicate the Bayesian networks that we need to perform conditional inference on, to compute correlations (top) and interventions (bottom). In a conditional query, we know from data that $\mathbf{x}_3 = \mathbf{s}_3$, and we infer the remaining variables by computing the posterior $p(\mathbf{x}_1, \mathbf{x}_2, \mathbf{x}_4 \mid \mathbf{x}_3 = \mathbf{s}_3)$. In an interventional query, we compute $p(\mathbf{x}_4 \mid do(\mathbf{x}_3 = \mathbf{s}_3))$. To do that, we have to first mutilate the structure of the graph, and then perform conditional inference on the new graph, with the joint probability $p(\mathbf{x}_1, \mathbf{x}_2, \mathbf{x}_3, \mathbf{x}_4) = p(\mathbf{x}_1)p(\mathbf{x}_2)p(\mathbf{x}_4 \mid \mathbf{x}_2, \mathbf{x}_3)$. More formally, consider the partition $\mathbf{X} = \mathbf{X}_{data} \cup \mathbf{X}_{unk}$, where $\mathbf{X}_{data} = \mathbf{x}_{i_1}, \ldots, \mathbf{x}_{i_n}$ is a subset of variables of which we have information via a data point $\mathbf{S}_{data} = \mathbf{s}_{i_1}, \ldots, \mathbf{s}_{i_n}$. We want to infer the values of the unknown nodes. This is trivial when a data point is the root of a tree only formed by unknown variables. In this case, it is possible to compute them via a forward pass. The problem, however, becomes more complex, and often intractable, when it is necessary to infer missing nodes that are parents of data points, as we need to invert the generative model of specific nodes. When dealing with continuous variables, we can use PC to perform such an inversion (Friston, 2005).

**Posterior distribution.** In most cases, the computation of the posterior distribution $p(\mathbf{X}_{unk} \mid \mathbf{X}_{data} = \mathbf{S}_{data})$ is intractable. A standard approach is to use variational inference with an approximate posterior $q(\mathbf{X}_{unk})$ restricted to belong to a family of distributions of simpler form than the true posterior. To make this approximation as similar as possible to the true posterior, the KL-divergence between the two distributions is minimized. Since the true posterior is not known, we instead minimize an upper bound on this KL divergence, known as the variational free energy:

$$F = \mathbb{E}_q[log(q(\mathbf{X}_{unk})) - \log(p(\mathbf{X}_{unk}, \mathbf{X}_{data}))].$$

We consider every edge $(i, j)$ to be a linear map $\mathbf{W}^{(i,j)}$ composed with a non-linearity $f(\mathbf{x})$ (such as ReLU). This defines how every parent node influences its child nodes. We further set the probability distribution of every node to be a multivariate Gaussian with unitary covariance matrix. In detail, every variable $\mathbf{x}_i$ is sampled from a Gaussian distribution of mean $\boldsymbol{\mu}_i$ and variance 1, where

$$\boldsymbol{\mu}_i = \sum_{k \in par(i)} \mathbf{W}^{(k,i)} f(\mathbf{x}_k). \tag{1}$$

To better derive a tractable formulation of the variational free energy, we use a mean-field approximation to assume a factorization into conditional independent terms, and assume that each of these terms is a Dirac delta (or, equivalently, a Laplace approximation). Note that these assumptions are standard in the literature (Friston, 2003; Friston et al., 2007; Millidge et al., 2021; Salvatori et al., 2022c), and lead to the following variational free energy:

$$F = \sum_i \|\mathbf{x}_i - \boldsymbol{\mu}_i\|^2 + ln(2\pi). \tag{2}$$

### 2.1 PREDICTIVE CODING GRAPHS

The derived variational free energy corresponds, up to an irrelevant constant, to the energy function of PC graphs, flexible models that can be queried in different ways (Salvatori et al., 2022a). Each vertex $v_i$ of a PC graph encodes several quantities: the main one is the value of its activity, which changes over time, and we refer to it as a *value node* $\mathbf{x}_{i,t}$. This is a parameter of the model, which is updated via gradient descent during inference. Additionally, each vertex has a *prediction* $\boldsymbol{\mu}_{i,t}$ of its value node, based on input from value nodes of other vertices, as detailed in Eq. 1. The *error* of every vertex at every time step $t$ is then given by the difference between its value node and its prediction, i.e., $\mathbf{e}_{i,t} = \mathbf{x}_{i,t} - \boldsymbol{\mu}_{i,t}$. This local definition of error allows PC graphs to learn using only local information. Here, we review the inference phase of PC, which computes correlations among data and results in an approximate Bayesian posterior over all node. It is also possible to train these models by updating the parameters $\mathbf{W}$ via stochastic gradient descent over a set of examples. For a detailed description of how *learning* on PC graphs work, we refer to Appendix A.

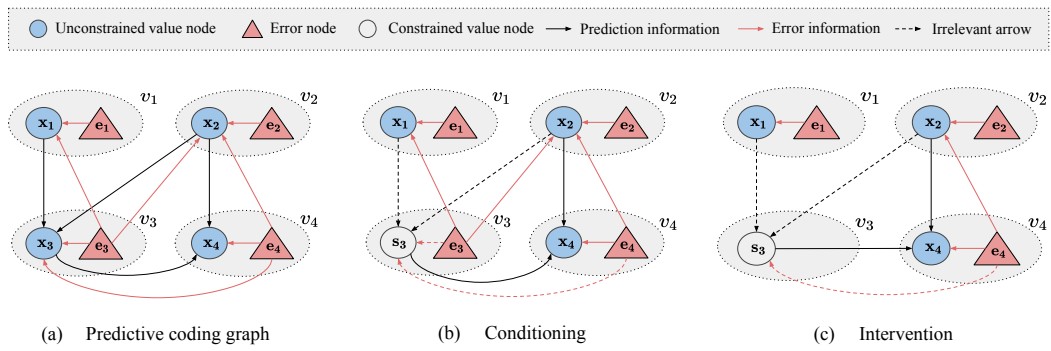

Figure 2: $(a)$ PC graph with the same causal structure of that in Fig. 1. Every vertex $v_i$ is associated with a value node $\mathbf{x}_i$, and an error node $\mathbf{e}_i$. The arrows show the influence of every node to the others: the prediction information follows the direction of the arrows of the original graph, while the error information goes backwards. $(b)$ Example of conditioning in PC graphs. We fix the value of $\mathbf{x}_3$, making the effect of all the arrows entering $v_3$ irrelevant, as $\mathbf{x}_3$ is fixed and hence ignores incoming information. This, however, does not apply to error information going *out* from $v_3$, which keeps influencing $\mathbf{x}_1$ and $\mathbf{x}_2$; this is solved in $(c)$ Example of an intervention in PC graphs. According to Pearl's causal theory, the do-operator on a node ($v_3$ in this case) removes the incoming edges, to avoid the newly introduced information to flow backwards and influence the parent nodes. As in PC the only information flowing opposite to the causal relations is the error information, an intervention can simply be performed by removing (or setting to zero) the error node.

**Query by conditioning.** Assume that we are presented with a data point $\mathbf{S}_{data} = \{\mathbf{s}_{i_1}, \ldots, \mathbf{s}_{i_n}\}$. Then, the value nodes $\mathbf{x}_{i_1}, \ldots, \mathbf{x}_{i_n}$ of the corresponding vertices are fixed to the entries $\mathbf{S}_{data}$ for every $t$, while the remaining ones are initialized to some random values, and continuously updated until convergence via gradient descent to minimize the energy function, following the rule $\Delta \mathbf{x}_{i,t} \propto \partial F_t / \partial \mathbf{x}_{i,t}$. The unconstrained sensory vertices will converge to a minimum of the energy given the fixed vertices, thus computing the conditional expectation of the latent vertices given the observed stimulus. Formally, the inference step estimates the conditional expectation

$$E(\mathbf{X}_T \mid \forall t: (\mathbf{x}_{i_1,t}, \ldots, \mathbf{x}_{i_n,t}) = (\mathbf{s}_{i_1}, \ldots, \mathbf{s}_{i_n})), \tag{3}$$

where $\mathbf{X}_T$ is the matrix of all value nodes at convergence. This computes the *correlation* among different parameters of the causal graph. In the next section, we show how to model *interventions* in PC graphs. For a neural implementation of a PC graph, see Fig. 2(a), and for the neural implementation of a conditional query, where the value of a specific node is fixed to a data point, see Fig. 2(b).

## 3 CAUSAL INFERENCE VIA PREDICTIVE CODING

The main goal of causal inference is to be able to simulate interventions in a process, and study the counterfactual effects of such interventions. In statistics, *interventions* are denoted by the $do(-)$ operator (Pearl, 1995; 2009). The value of a random variable $\mathbf{x}_i$ when performing an intervention on a different variable $\mathbf{x}_j$ is denoted by $p(\mathbf{x}_i \mid do(\mathbf{x}_j = \mathbf{s}))$. This is equivalent to the question *What would $\mathbf{x}_i$ be in this environment if we set $\mathbf{x}_j = \mathbf{s}$?* In the case of the example in Fig. 1, the question could be *What would the expected income level be, if we change the education level of this person?* In fact, while 'education' and 'income level' may be correlated by a hidden confounder (intelligence, in this case), an intervention removes this correlation by changing the education level of a randomly selected individual, regardless of level of intelligence. To perform an intervention on a Bayesian network, we first have to act on the structure of the graph, and then query the model by conditioning on the new graph, as shown in Fig. 1. Assume that we have a graph $G$, and we want to know the value of $\mathbf{x}_i$ after performing an intervention on $\mathbf{x}_j$ by fixing it to some value $s$. This can be done according to the two following steps:

1. Generate a new graph $G'$ by removing all the in-coming edges of $v_i$ from $G$.
2. Compute the conditional expectation $E(\mathbf{X} \mid \mathbf{x}_j = \mathbf{s})$ using $G'$.

**Interventional query.** In a PC graph, the only information that flows in the opposite direction of an arrow is the prediction error. In fact, if we have $v_1 \to v_2$, the update of the value node $\mathbf{x}_1$ is

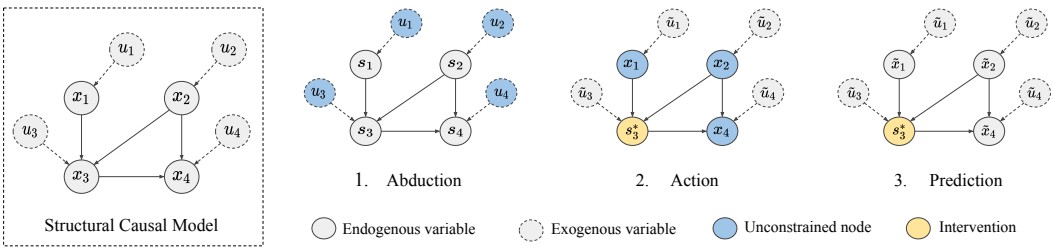

Figure 3: *What would $\mathbf{x}_4$ be, had $\mathbf{x}_3$ been equal to $\mathbf{s}_3^*$ in situation $U = \mathbf{u}$?* This figure provides an example of the three-step process to perform counterfactuals, using a structural causal model with four exogenous and four endogenous variables. We are given two kinds of data: the original values of $\mathbf{x}_1, \ldots, \mathbf{x}_4$, which correspond to past information, here denoted by $\mathbf{s}_1, \ldots, \mathbf{s}_4$, and the intervention information $\mathbf{s}_3^*$, needed to understand the *what would have happened to $\mathbf{x}_4$ if we had changed $\mathbf{s}_3$ to $\mathbf{s}_3^*$?*. The final answer corresponds to the node $\tilde{\mathbf{x}}_4$ obtained in the prediction step.

affected by the error $\mathbf{e}_2$. To avoid this and perform an intervention, we set the value of $\mathbf{e}_2$ to zero throughout the inference phase. This is convenient, since it allows us to not directly act on the structure of the graph to perform interventions but rather perform them dynamically 'at runtime', which results in increased efficiency in the case of nodes with numerous incoming edges. We assume hard interventions over soft ones (Correa & Bareinboim, 2020), thus eliminating all parent variable effects. This preserves the classical 3-step procedure for computing counterfactuals. The key distinction is how interventions are executed on the PC graph, specifically by nullifying the prediction error. This approach obviates the need for explicit adjustment formulas and back-door criteria in causal inference. Hence, we have the following theorem, proven in Appendix B:

**Theorem 3.1.** *Let $\mathcal{G}$ be a PC graph with structure given by a directed acyclic graph $G$, and let us assume we perform an intervention $do(\mathbf{x}_j) = \mathbf{s}$ via a conditional query on the mutilated PC graph. The distribution of the variables obtained is equivalent to the one obtained by running inference on the original PC graph while setting both $\mathbf{x}_{j,t} = \mathbf{s}$ and $\mathbf{e}_{j,t} = 0$ for every $t > 0$. That is,*

$$E(\mathbf{X_T} \mid do(\mathbf{x}_j = \mathbf{s})) = E(\mathbf{X_T} \mid \forall t : \mathbf{x}_{j,t} = \mathbf{s}, \mathbf{e}_{j,t} = 0 \,\forall t). \tag{4}$$

### 3.1 STRUCTURAL CAUSAL MODELS

While interventions serve to answer questions about the consequences of actions performed in the present, counterfactuals are used to study interventions in the past. For instance, we could ask: *What would the value of $\mathbf{x}_i$ have been if $\mathbf{x}_j$ had been set to $\mathbf{s}_j^*$, given a particular context $U = \mathbf{u}$?* Using a concrete example, in reference to Fig. 1: *What would this person's income have been if they had earned a master's degree, under the conditions defined by $U = \mathbf{u}$?* This is modeled using Structural Causal Models (SCMs). A SCM is a triple $(U, V, F)$, where $V$ is the set of endogenous (observable) variables corresponding to the internal vertices of the causal graph, $U$ is the set of exogenous (unobservable) variables that serve as root nodes in the graph, and $F$ is the set of functions that determine the values of endogenous variables according to the structure of $G$. An example of a SCM is represented in Fig. 3. Then, counterfactual inference with an SCM involves three steps:

1. **Abduction**: Here, we are provided with the values $(\mathbf{s}_1, \ldots, \mathbf{s}_N)$ of the endogenous nodes in $V$. We use them to compute the values of the exogenous variables, which we denote by $\tilde{u}_1, \ldots, \tilde{u}_N$. Hence, according to the following:

$$E(\mathbf{u}_1, \ldots, \mathbf{u}_N \mid \forall t : (\mathbf{x}_1, \ldots, \mathbf{x}_N) = (\mathbf{s}_1, \ldots, \mathbf{s}_N)). \tag{5}$$

2. **Action**: Now that we computed the values of the exogenous variables, we fix them and perform an intervention on $\mathbf{x}_j$. Particularly, we set $\mathbf{x}_j = \mathbf{s}_j^*$, and we set $\mathbf{e}_j = 0$, which has the effect of removing any influence of $x_j$ on its parent nodes.

3. **Prediction**: We now have all the elements to compute the counterfactual on $\mathbf{x}_i$, which is:

$$E(\mathbf{x}_i \mid \forall t : (\mathbf{u}_1, \ldots, \mathbf{u}_M) = (\tilde{\mathbf{u}}_1, \ldots, \tilde{\mathbf{u}}_M), \mathbf{x}_j = \mathbf{s}_j^*, \mathbf{e}_j = 0). \tag{6}$$

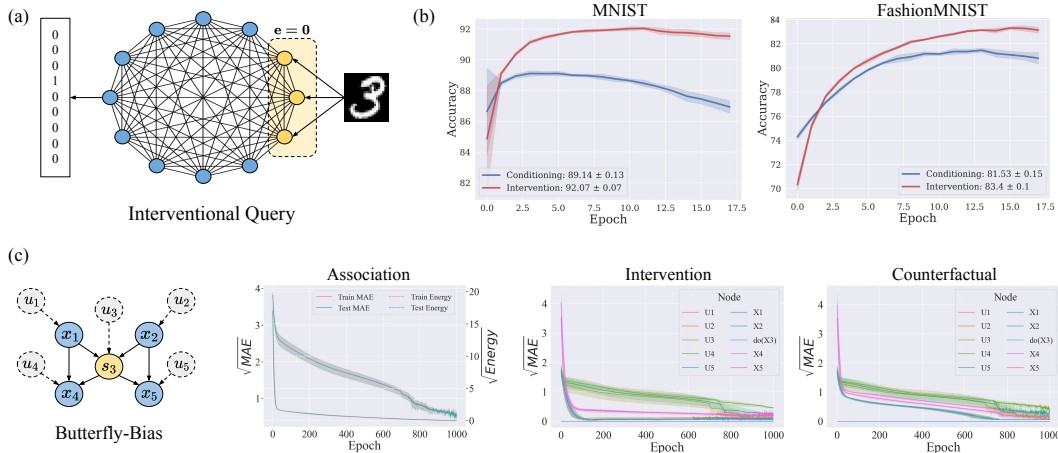

Figure 4: (a) How to compute a prediction given a data point on a fully connected PC graph, using interventional queries. (b) Test accuracy over epochs computed via query by conditioning and query by intervention. (c) Left to right: causal structure of the SCM. Convergence behavior of PC energy vs. error metric (MAE), during SCM learning for butterfly graph. Error (by node) of interventional query estimates on $\mathbf{x}_3$ (yellow node). Error (by node) of counterfactual query estimates with intervention on $\mathbf{x}_3$ given factual data (blue nodes).

## 3.2 EXPERIMENTS

We now perform experiments that confirm the technical discussion and claims made in the previous section. To this end, we test PC graphs in their ability to compute causal queries on three levels of Pearl's ladder of causation (2009), namely, association (level 1), intervention (level 2), and counterfactual (at level 3) for both linear and non-linear data. Then, we show how interventions can be used to improve the performance of classification tasks on fully connected models. We conclude with an experiment showcasing the robustness of PC graphs and their learned representations of complex data, specifically, with respect to counterfactual interventions on distinct attributes of images. We assume causal sufficiency (no unobserved confounders) and compare against works that make this assumption. The appendix contains extensive results and details to reproduce our approach.

**Causal Inference.** We evaluate associational, interventional, and counterfactual queries on various causal graphs (Fig. 7) using linear and non-linear data. We assume that known graphs and the SCMs with additive Gaussian noise are unspecified. We generate synthetic observational, interventional, and counterfactual data for testing. The observational data are generated by randomly sampling values for the exogenous variables, $\boldsymbol{\mu}$. We then use the exogenous values to compute the values of the endogenous variables $\mathbf{x}$. Interventional data are similarly generated, but the structural equations are altered by an intervention. Finally, the counterfactual data consist of pairs, $(\mathbf{x}, \mathbf{x}')$ with $\mathbf{x}$ being observational and $\mathbf{x}'$ interventional data, both sharing the same $\boldsymbol{\mu}$. To perform causal inference, we fit the PC graph to the observed data to learn the parameters of the structural equations, including proxy noise distribution parameters. We evaluate the learned SCM by comparing various difference metrics between true and inferred counterfactual values. Details on all metrics are in Appendix C.

Here, we only provide results for the most interesting and complex graph among the proposed ones, namely, the butterfly, represented in Fig. 4(c). In Appendix C, we provide a detailed study of all the aforementioned graphs, on a large number of metrics. The results show that PC graphs accurately estimate causal queries for linear and more complex non-linear data. Our method outperforms state-of-the-art methods like CAREFL (Khemakhem et al., 2021), VACA (Sánchez-Martin et al., 2022), and MultiCVAE (Karimi et al., 2020) by large margins, while at the same time requiring significantly fewer parameters. The plots in Fig. 4(c) show that the model is able to correctly infer interventional and counterfactual queries, as shown by the converging MAE of non-intervened nodes. Finally, unlike (Khemakhem et al., 2021), we do not reduce the graph to its causal ordering and the performance of PC graphs remains stable as causal paths get longer, an issue seen in (Sánchez-Martin et al., 2022).

**Classification.**    In the original work on PC graphs (Salvatori et al., 2022a), the authors have trained a fully connected model to perform classification tasks using conditional queries. The performances are poor compared to those of hierarchical models for two reasons: first, conditional queries do not impose any direction to the information flow, making the graph learn $p(\mathbf{y}|\mathbf{x})$ as well as $p(\mathbf{x}|\mathbf{y})$, even though we only need the first term. Similarly, the model also learns $P(\mathbf{x})$ and $P(\mathbf{y})$, which is, how the prior depends on itself. Second, the model complexity is too high, not allowing the model to use any kind of background knowledge, such as hierarchical/structural information/prior, usually present in sparser models. Here, we address the first limitation by performing an intervention on the input: this prevents the error of the inputs to spread in the network, and enforces a specific direction of the information flow, which goes from cause (the image) to effect (the label). To assess the impact of such an intervention on the test accuracy, we train a fully connected model with 2000 neurons on the MNIST and FashionMNIST datasets, and compute the test accuracy for conditional and interventional queries. We perform a large hyperparameter search on learning rates, activation functions, and weight decay values. In all cases, performing an intervention improves the results. In Fig. 6(d), we present an example showcasing the best results obtained after hyperparameter search. The interventional query led to improvements in the final test accuracy of almost $2\%$ for both datasets. The experiment details can be found in Appendix E.

**Robustness.**    A recent work demonstrates that existing deep-learning methods fail to obtain sufficient robustness or performance on counterfactual queries in certain scenarios (De Brouwer, 2022). We show that PC surpasses current state-of-the-art results for counterfactual queries while requiring a simpler architecture, and without relying on ad hoc training techniques. We evaluate the robustness of our model on counterfactual inference tasks of higher dimensions, thereby examining the feasibility of our method to perform causal inference on more complex data. The dataset we consider consists of tuples $(\mathbf{x}, \mathbf{u}_z, \mathcal{T}, \mathbf{y}, \mathcal{T}', \mathbf{y}')$, where $\mathbf{x}$ is an image from the MNIST dataset, $\mathcal{T}$ is the assigned treatment, which is a rotation angle (confounded by $\mathbf{x}$). Furthermore, $\mathbf{u}_z$ is a hidden exogenous random variable that determines the color of the observed outcome image $\mathbf{y}$, that is added to the forth variable $\mathbf{y}'$, a colored and rotated MNIST image representing the counterfactual response obtained when applying the alternative treatment $\mathcal{T}'$. We consider SCMs with 4 nodes that encode the four variables, as sketched in Fig. 5. Here, every edge represents a feed-forward network of different depth and hidden dimension 1024. A detailed explanation of how to reproduce the results is given in Appendix F.

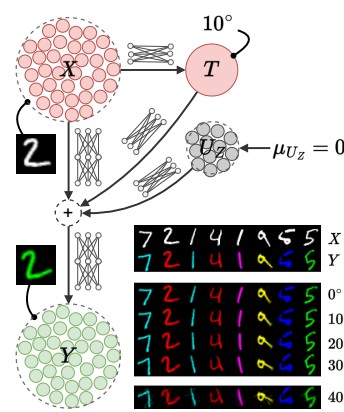

Figure 5: Architecture used to reconstruct counterfactual images. $U_Z$ corresponds to the color of the digit, $T$ to the rotation angle, $X$ to the input, $Y$ to the colored and rotated image. The colored digits show that our method is robust when performing interventions on the rotation angle.

**Results.**    The results show that PC graphs improve on state-of-the-art methods, despite the fact that we do not use convolutional layers like in the original work. First, the generated images have an MSE on the test set $(0.0008 \pm 0.0002)$ that is lower than that reported in the original work $(0.001 \pm 0.001)$. The high quality of reconstruction is also visible in the generated images reported in Fig. 5. Compared to the work (De Brouwer, 2022), we are able to generalize to rotations of $40°$ (absent in the training data), even if this introduces some noise in the generated output. Furthermore, contrary to the original model, our architecture is robust with respect to the choice of the hyperparameter linked to $u_z$ and does not necessitate to perform a hyperparameter sweep to find the right value. Hence, we conclude that PC graphs are able to correctly model the treatment rotation in the counterfactual outcome, while keeping the color, which is independent of rotation, unchanged.

## 4    STRUCTURE LEARNING

Learning the underlying causal structure from observational data is a critical and highly active research area, primarily due to its implications for explainability and modeling interventions. Traditional approaches use combinatorial search algorithms to find causal structures. However, such methods tend to become computationally expensive and slow as the complexity (e.g., the number of nodes) of the graph increases, as shown in previous works (Chickering, 1996; 2002). Therefore, we

focus on gradient-based learning methods instead (Zheng et al., 2018), as these allow us to handle larger causal graph structures in a computationally efficient manner. Let us consider $\mathbf{A}$ to be the adjacency matrix of a graph. Ideally, this matrix should be a binary matrix with the property that $a_{i,j} = 1$, if there exists an edge from $v_i$ to $v_j$, and 0, otherwise. From a Bayesian perspective, our method learns the marginal of the graph edges, where $\mathbf{A}$ is a matrix composed of continuous, learnable parameters, which assign weights to signify importance of specific connections. To this end, we can consider every PC graph to be fully connected, where the prediction of every node $\mathbf{x}_i$ now depends on the entries of the adjacency matrix:

$$\boldsymbol{\mu}_i = \sum_{k=0}^{N} a_{k,i} f_{k,i}(\mathbf{x}_k), \qquad \text{update rule}: \ \Delta a_{i,j} \propto -\frac{\partial F}{\partial a_{i,j}} = \beta \cdot \mathbf{e}_{i,T} \mathbf{W}^\top f(\mathbf{x}_{j,T}), \qquad (7)$$

where $\beta$ is the learning rate of the parameters $a_{i,j}$. Then, the entries of $\mathbf{A}$ are updated via gradient descent to minimize the variational free energy of Eq. 2. Our goal is to learn an acyclic, sparsely connected graph, which requires a prior distribution that enforces these two constraints. We consider three possible priors: a Gaussian prior, a Laplace prior, and the acyclicity prior. The latter prior is equal to zero if and only if the corresponding graph is acyclic (Zheng et al., 2018):

$$l(\mathbf{A}) = exp(-\sum_{i,j} |a_{i,j}|), \qquad g(\mathbf{A}) = \mathcal{N}(0,1), \qquad h(\mathbf{A}) = tr(exp(\mathbf{A} \times \mathbf{A})) - d.$$

The energy function that we aim to minimize via gradient descent is given by the sum of the total energy, as defined in Eq. 2, and the three aforementioned prior distributions, each weighted by a scaling coefficient. The first two priors effectively apply the $L1$ and $L2$ norms to the parameters of the adjacency matrix, and they form the elastic norm when used in conjunction.

**Negative Examples.** The regulariser $h(A)$ introduces an inductive bias that may be undesirable, as we know that cyclic structures may be beneficial in several tasks (Salvatori et al., 2022a). Without $h(A)$, however, training converges towards a degenerate structure, as shown on the top right of Fig. 6(c), where each output node predicts itself, ignoring any contribution of the input nodes. We solve this degeneration behavior by introducing negative examples, which are data points with a wrong label, into the training set. The model is then trained in a contrastive way (Chen et al., 2020), i.e., by *increasing* the prediction error of every node for negative examples $k$, and decreasing it otherwise, when the label is correct, as shown in Fig. 6(c). A detailed explanation of how training with negative samples works is given in Appendix G. We show that negative examples address the convergence issue towards a degenerate graph by rendering the label nodes contingent on the inputs, thus steering the model towards adopting a hierarchical structure instead.

## 4.1 EXPERIMENTS

We perform two different structure learning experiments. In the first, we assume that we are provided with non-interventional data generated by from a Bayesian network of unknown structure. The task is to retrieve the original graph starting with a fully connected PC graph, which is a standard problem in causal discovery (Morales-Alvarez et al., 2022; Geffner et al., 2022; Zheng et al., 2018). In the second experiment, we perform classification for MNIST and FashionMNIST using a fully connected PC graph. This time, however, we augment the classification objective with priors to enforce sparsity and acyclicity, and conjecture that $(i)$ this improves the final test accuracy, and $(ii)$ reduces a fully connected graph to the "correct" sparse network, i.e., the hierarchical one.

**Structure Learning** Here, we use synthetic data sampled from an SCM with $N \in \{10, 15, 20\}$ nodes and $\{1, 2, 4\}$ expected edges per node. The graph structure is either an Erdős-Rényi or a scale-free random graph. To this end, we denote an Erdős-Rényi graph with $2N$ expected edges as ER2 or a scale-free graph with $4N$ expected edges as SF4, respectively. We vary the graph type, number of nodes and/or edges, to test the scalability and stability of each method. We place uniformly random edge weights onto a binary adjacency matrix of a graph, to obtain a weighted adjacency matrix, $\mathbf{W}$. We sample observational data from a set of linear structural equations with additive Gaussian noise.

Due to the linearity, we model each edge as a scalar. Hence, we can set the parameters of the weighted adjacency matrix to be the estimated model parameters $\widehat{\mathbf{W}}$. To prune the parameters that are not used in the linear structural equations that generate the observed data, we require our model to be sparse and acyclic. Thus, we consider the parameters to have prior distributions $h(\mathbf{W})$ and $l(\mathbf{W})$. Then, the experiment consists of training the fully connected model to fit the dataset, and checking whether the PC graph can converge to the random graph structure that generated the data.

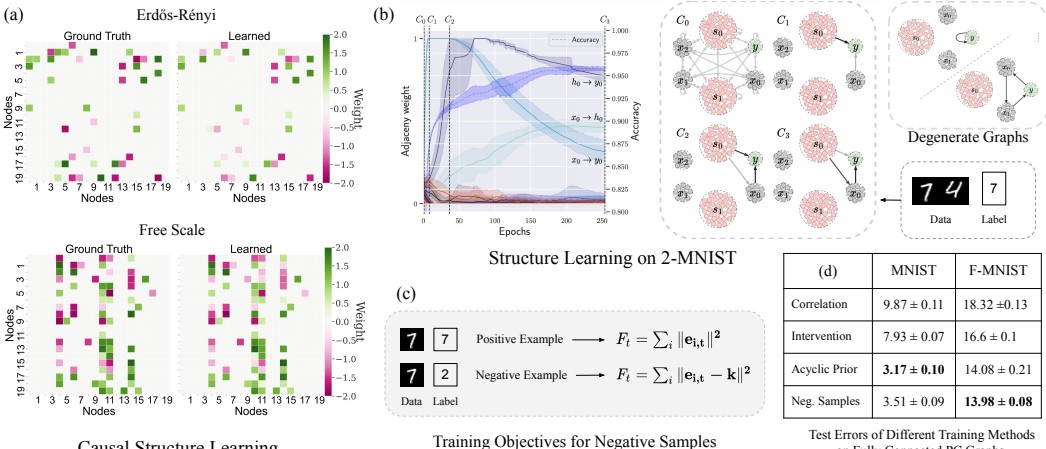

Figure 6: (a) Experiments on structure learning from synthetic data, generated from Erdős-Rényi and scale-free random graphs with 20 nodes. On the left, the connection strength of the true graph; on the right, the one learned by a PC graph. (b) Structure learning on the 2-MNIST dataset: the plot shows the weights of the adjacency matrix $\mathbf{A}$ over the number of epochs, the sketch on the right the resulting PC graph. (c) A description of the two energy functions optimized by the PC graph when training on negative and non-negative examples. (d) Table with test error of all experiments performed on MNIST and FashionMNIST, averaged over three seeds. The best results are obtained when augmenting the training process with both the proposed structure learning methods.

The results show that PC graphs are able to infer the structure of the data generating process for arbitrary dense random graphs of various complexities. The heatmaps in Fig. 6(a) show that our method can estimate the true weight adjacency matrix for dense SF4 and ER2 graphs with 20 nodes. Hence, we conclude that the learned adjacency matrix, which we chose as the median performing model across multiple seeds, is able to well capture the variable dependencies of the observed data. More details on the dataset generation, how the experiment is performed, quantitative results for various structure learning metrics, and detailed comparisons against highly used baseline methods (PC (Kalisch & Bühlman, 2007), GES (Chickering, 2002), NOTEARS (Zheng et al., 2018), and ICALiNGAM (Shimizu et al., 2006)), are provided in Appendix G. In contrast to the baselines, which all perform worse than our method, our algorithm sustains a stable performance in all ER and SF graph setups, as validated by structural Hamming distance (SHD), F1 score, and other metrics.

**Classification.** Here, we extend the study on the experiments performed in Section 3, and check whether we are able to improve the results by allowing the PC graph to cut extra connections during training. Additionally, we create a new dataset, called 2-MNIST, whose data consist of pairs $(\mathbf{s}_0, \mathbf{s}_1)$ of MNIST images, and the label is the label of $\mathbf{s}_0$. This is to check whether PC networks are able to understand the underlying causal structure of the dataset, and remove connections that start from $\mathbf{s}_1$. As an architecture, we consider a PC graph with 6 nodes, one of dimension 784, one of dimension 10, and 4 hidden nodes of dimension $d$. In the case of 2-MNIST, we have two nodes of dimension 784, and only three of dimension $d$. The adjacency matrix $\mathbf{A}$ has then dimension $6 \times 6$. Note that, when the entries of $\mathbf{A}$ are all equal to one, then this model is equivalent to the fully connected one of Section 3. Here, however, we propose two techniques to augment the training process, and let the model converge to a hierarchical network. The first one consists of adding the three proposed priors on the matrix $\mathbf{A}$, to enforce sparsity and acyclicity in the graph; the second one consists of augmenting the dataset via negative examples, while enforcing sparsity via the Laplace prior only. Note that enforcing acyclicity is fundamental, otherwise the circular dependencies in the graph would make the model converge to degenerate structures, such as the ones provided in the top right corner of Fig. 6. More details on this can be found in Appendix E.

In the first experiment, we use the 2-MNIST dataset to test whether the acyclic and sparse priors are able to both remove the out-going connections from the second image $\mathbf{s}_2$, and learn a hierarchical structure, which we know to be the best one to perform classification on MNIST. In the second experiment, we train the same fully-connected model of Section 3 and check whether the priors allow

to increase the classification accuracy of the model. To conclude, we perform a classification task with the negative examples and the Laplace prior, to test whether this method also allows to avoid converging to degenerated graph structures. The results on the 2-MNIST dataset show that the model immediately prunes the parameters out-going from $s_2$. In the first 100 epochs, the edge with the largest weight is the linear one, which directly connects the input to the label. While this shows that the model correctly learned the dependencies, linear classification on MNIST and FashionMNIST does not yield great accuracies. This problem is naturally addressed in the later stages of the training process, where the entry of the adjacency matrix relative to the linear map loses weights, and hence influence on the final performance of the model. When training finally converges, the resulting model is hierarchical, with one hidden layer, as shown in the plot in Fig. 6(b). This shows that PC graphs are not only able to learn the causal dependencies correctly, but also to be able to discriminate among these structures, and converge to a well performing one.

In the second experiment, where we perform classification on MNIST and FashionMNIST with $h(\mathbf{A})$, the model shows a clear improvement over the baseline proposed in Section 3. The same applies for the training with negative examples, as reported in the table in Fig. 6(d), which shows a performance comparable to these of training with an acyclicity prior. To reach the usual results that can be obtained via standard neural networks trained with backpropagation (i.e., a test error $< 2\%$), it suffices to fine-tune the model using the newly learned structure.

## 5 RELATED WORK

In the last years, there have been numerous works that have tackled machine learning problems using PC networks. They have been shown to perform well in classification tasks using all kinds of architectures, such as feedforward and convolutional models, graph neural networks, and transformers (Whittington & Bogacz, 2017; Han et al., 2018; Salvatori et al., 2022c; Byiringiro et al., 2022; Pinchetti et al., 2022). These results are partially justified by some similarities that PC shares with backprop when performing supervised learning (Song et al., 2020; Millidge et al., 2020; Salvatori et al., 2022b). Multiple works have also applied it to tasks such as image generation (Ororbia & Kifer, 2022; Ororbia & Mali, 2019), continual learning (Ororbia et al., 2022; Song et al., 2022), and associative memories (Salvatori et al., 2021; Yoo & Wood, 2022; Tang et al., 2023).

Causality has found applications in problems such as treatment effect estimation, time series modeling, image generation, and natural language processing, as well as enhancing interpretability and fairness in machine learning (Shalit et al., 2017; Runge et al., 2019; Lopez-Paz et al., 2017; Kaushik et al., 2019; Kusner et al., 2017). Different works have used deep generative modeling techniques to investigate causality problems, such as Graph Neural Networks, Variational Autoencoders, flow and diffusion models (Sanchez & Tsaftaris, 2022; Khemakhem et al., 2021; Karimi et al., 2020; Pawlowski et al., 2020; Yu et al., 2019). Some works study the problem of learning the causal structure from observational data, previously done via combinatorial search (Spirtes et al., 2000; Chickering, 2002; Shimizu et al., 2006; Kalisch & Bühlman, 2007). However, combinatorial searches algorithm grow double exponentially in complexity with respect to the dimension of the graph. To this end, recent works mostly performing continuous optimization by using the acyclic prior we have also discussed in our work (Zheng et al., 2018; Bello et al., 2022; Yu et al., 2019).

## 6 CONCLUSION

We have provided a bridge between the fields of causality and computational neuroscience by showing that predictive coding graphs have the ability of both learning the DAG structures from observational data, and modeling associational, interventional, and counterfactual distributions (Geffner et al., 2022; Sharma & Kiciman, 2020). This makes our method suitable candidate for an end-to-end causality engine, which can answer causal queries without knowing detailed structural equations of an SCM. In the case of causal inference, we have shown how interventions can be performed by setting prediction errors of nodes that we are intervening on to zero, and how this leads to the formulation of predictive-coding based structural causal models that go beyond correlations. For structure learning, we have shown how to use existing techniques to derive causal relations from observational data. More generally, this work further highlights the flexibility of predictive coding models, which can be used to both train deep neural networks that perform well on different machine learning tasks, and to perform causal inference on directed graphical models, extending the toolbox of computational neuroscientists allowing them to compute causal queries and structure learning.

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

## REPRODUCIBILITY STATEMENT

In the appendix, we offer detailed algorithmic descriptions, pseudocode, and an exhaustive list of hyperparameters and models. This supports our claims and provides a rigorous comparison with established baselines. We have rigorously verified our implementations through consistent results, achieved by setting random seeds and conducting multiple experimental runs. All experiments utilized a machine with an Intel Core CPU and NVIDIA RTX3090 GPU. Unless otherwise specified, we assume causal sufficiency to eliminate the possibility of unobserved confounding. Our algorithms recover causal structures up to a Markov equivalence class (MEC). We exclusively employ hard interventions for handling both interventional and counterfactual queries. This comprehensive documentation aims to facilitate both the reproducibility and understanding of our methodology.

## ETHICS STATEMENT

In the domain of causal structure learning from observational data, it's crucial to recognize that algorithms can only identify the causal graph within the confines of a Markov Equivalence Class (MEC). This limitation means that various causal graphs, each with differing implications, could be equally supported by the observational data. Our approach does not incorporate interventional data, amplifying the need for caution in its application for causal discovery. The method should be used alongside further assumptions and in consultation with subject-matter experts to minimize the risk of affirming false causal relationships. Incorrect usage has the potential for negative societal consequences, such as poor or biased decision-making processes.

