CONTENTS

---

**Algorithm 1** Learning a datapoint $\mathbf{S}_{data} = \mathbf{s}_{i_1}, \ldots, \mathbf{s}_{i_n}$

---

**Require:** $(\mathbf{x}_{i_1,t}, \ldots, \mathbf{x}_{i_n,t})$ is fixed to $(\mathbf{s}_{i_1}, \ldots, \mathbf{s}_{i_n})$ for every $t$.
1: **for** $t = 1$ to $T$ **do**
2:     **for** each vertex $i$ **do**
3:         update $\mathbf{x}_{i,t}$ to minimize $F_t$ via Eq. equation 8
4:     **end for**
5:     **if** $t = T$ **then**
6:         update every $\mathbf{W}_{i,j}$ to minimize $F_t$ via Eq. equation 9.
           update every $a_{i,j}$ to minimize $F_t$ via Eq. equation 10.
7:     **end if**
8: **end for**

---

# Appendix

## A   Learning on PC Graphs

Given a labeled point, two phases are needed to perform a single weight update. The first one, called *inference*, is used both in the training phase, to compute the best configuration of value nodes to perform a weight update, and in the prediction phase, to compute an output when provided a specific input. The inference phase corresponds to *Query by conditioning*, as described in Section 2. During this phase, the weights are frozen, and only the internal value nodes are updated to minimize the free energy. The second phase happens after the inference has converged, and hence the 'best' neural activities are computed. Here, the opposite happens: all the value nodes are now frozen, and a single weight update is performed to further minimize the same energy function. If we are considering models with an adjacency matrix $A$, we also update its parameters. We will now provide a more formal description of the two phases.

Let us assume we are presented with a datapoint $\mathbf{S}_{data} = \mathbf{s}_{i_1}, \ldots, \mathbf{s}_{i_n}$. First, the value nodes of the vertices $v_{i_1}, \ldots, v_{i_n}$ are fixed to be equal to the entries of $\mathbf{S}_{data}$ for the whole duration of the training process, i.e., for every $t$. Second, the variational free energy is minimized via gradient descent on the value nodes, During this phase, the weights are fixed, and the value nodes are updated as follows:

$$\Delta\mathbf{x}_{i,t} = -\gamma\frac{\partial F_t}{\partial\mathbf{x}_{i,t}} = \gamma \cdot (-\mathbf{e}_{i,t} + f'(\mathbf{x}_{i,t})\textstyle\sum_{k\in ch(i)}\mathbf{e}_{k,t}\mathbf{W}_{i,k}), \tag{8}$$

where $\gamma$ is the a positive real number that indicates a learning rate. When the inference phase is completed, the value nodes get fixed, and a single weight update is performed as follows:

$$\Delta\mathbf{W}_{i,j} = -\alpha \cdot \frac{\partial F_t}{\partial\mathbf{W}_{i,j}} = \alpha \cdot \mathbf{e}_{i,T}f(\mathbf{x}_{j,T}). \tag{9}$$

To conclude, the update of the entries of the adjacency matrix (without the priors), are the following:

$$\Delta a_{i,j} = -\beta\frac{\partial F_t}{\partial a_{i,j}} = \beta \cdot \mathbf{e}_{i,T}\mathbf{W}^\top f(\mathbf{x}_{j,T}). \tag{10}$$

We provide the pseudocode of the training process on PC graphs in Algorithm 1.

## B   Proof of Theorem 1

*Proof of Theorem 1.* We seek to prove:

$$E(\mathbf{x}_j \mid do(\mathbf{x}_i = \mathbf{s})) = E(\mathbf{x}_{j,T} \mid \forall t : \mathbf{x}_{i,t} = \mathbf{s}, \mathbf{e}_{i,t} = 0).$$

Let $G'$ be the mutilated graph structure of the Bayesian network $G$ after a do-operation. Then, by definition of the expectation of interventional distributions, we have that

$$E(\mathbf{x}_j \mid do(\mathbf{x}_i = \mathbf{s}))_G = E(\mathbf{x}_j \mid \mathbf{x}_i = \mathbf{s})_{G'},$$

where the expectations are computed, respectively, on graph $G$ and $G'$. We utilize the value node update rule for $\Delta \mathbf{x}_{i,t}$ defined in Eq. 8. Our aim is to demonstrate that the node values in the PC graphs defined on $G$ (with $\mathbf{e}_i = 0$) and $G'$ follow the same update dynamics and thus have identical distributions.

**Case 1: Parents of the Intervention Node $x_i$**

In $G$, the value node update rule for any parent $\mathbf{x}_j$ of $x_i$ is:

$$\Delta \mathbf{x}_{j,t}^G = \gamma \cdot (-\mathbf{e}_{j,t} + f'(\mathbf{x}_{j,t}) \sum_{k \in ch(j)} \mathbf{e}_{k,t} \mathbf{W}_{j,k}).$$

When $\mathbf{e}_i = 0$, the term involving $\mathbf{e}_i$ is omitted, yielding:

$$\Delta \mathbf{x}_{j,t}^G = \gamma \cdot (-\mathbf{e}_{j,t} + f'(\mathbf{x}_{j,t}) \sum_{k \in ch(j) \setminus \{i\}} \mathbf{e}_{k,t} \mathbf{W}_{j,k}).$$

In $G'$, $\mathbf{x}_i$ is removed due to the do-operation, resulting in an identical update rule:

$$\Delta \mathbf{x}_{j,t}^{G'} = \Delta \mathbf{x}_{j,t}^G.$$

**Case 2: The Intervention Node $\mathbf{x}_i$ Itself**

In both $G$ and $G'$, the value $x_i$ remains constant at $\mathbf{s}$, making its update rule irrelevant.

**Case 3: Children of the Intervention Node $x_i$**

For a child $x_j$ of $x_i$ in $G$, the update rule is:

$$\Delta \mathbf{x}_{j,t}^G = \gamma \cdot (-\mathbf{e}_{j,t} + f'(\mathbf{x}_{j,t}) \sum_{k \in ch(j)} \mathbf{e}_{k,t} \mathbf{W}_{j,k}).$$

This update rule remains unchanged in $G'$:

$$\Delta \mathbf{x}_{j,t}^{G'} = \Delta \mathbf{x}_{j,t}^G.$$

By addressing all three cases, we show that value nodes in $G$ (with $e_i = 0$) and $G'$ follow identical update dynamics. Therefore, the distributions of remaining variables in $G'$ and $G$ (with $\mathbf{e}_i = 0$) are the same, completing the proof. $\qquad\square$

## C    Interventional and Counterfactual Inference

Here, we provide a detailed discussion on the experiments proposed in Section 3, where we test the ability of PC graphs to model interventional and counterfactual queries. The core of decision making is to be able to determine which intervention/action results in an outcome of interest. As such, being able to answer causal queries on a variety of DAG structures and intervention nodes in a DAG is essential. The causal inference approach that we propose only requires knowledge of the causal structure in the form of parent-child relationships among endogenous variables $\mathbf{x}$. We assume the parameters of the structural equations, $F$, to be unknown. We make the causal sufficiency assumption, meaning that there is no hidden confounding (Peters et al., 2017). This is achieved by having an independent exogenous variable for each endogenous variable.

**Setup.** We test the associational, interventional, and counterfactual query capabilities of our method on different common causal graph structures with $N$ endogenous nodes, namely (i) collider ($N = 3$), (ii) confounder ($N = 3$), (iii) mediator ($N = 3$), (iv) chain ($N = 3$), (v) fork ($N = 3$), (vi) M-bias ($N = 5$), and (vii) butterfly bias ($N = 5$). Each of these structures is visualized in Fig. 7. For every structure, we generate datasets from a linear SCM with additive Gaussian noise and no restrictions on the location and scale parameters. We use observational training data, $\mathbf{X}$, to fit the PC model. We learn each structural equation, $F_i$, via a training scheme in which we infer the exogenous variables, $\mathbf{u}$, given observed endogenous variables, $\mathbf{x}$, from the training dataset. As such, the SCM is fitted by estimating the parameters of each exogenous variable according to $\mathbf{u}_i \sim \mathcal{N}(\boldsymbol{\mu}_i, \sigma_i)$. This way, we learn to approximate the distribution of the SCM's exogenous variables $\mathbf{u}$. Note that during the training process, we do not use the factual data once the abduction step

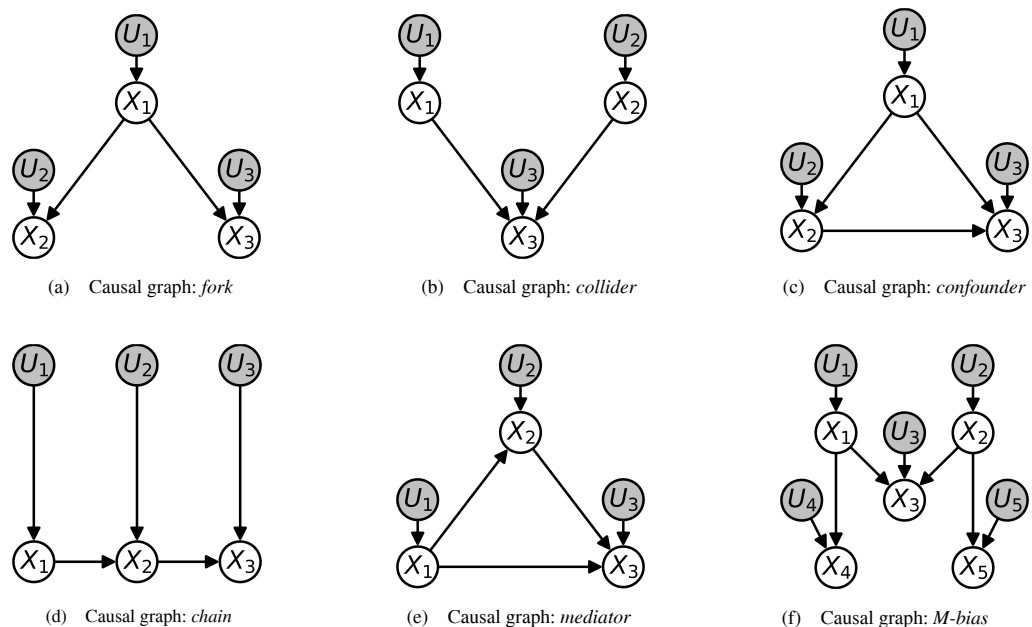

(a)   Causal graph: *fork*

(b)   Causal graph: *collider*

(c)   Causal graph: *confounder*

(d)   Causal graph: *chain*

(e)   Causal graph: *mediator*

(f)   Causal graph: *M-bias*

Figure 7: Additional graph structures used in experiments of Section 3. White nodes are endogenous variables and observed. Shaded nodes denote independent, exogenous variables for which we do not observe data unless they have no parents in which case $\mathbf{x}_i \coloneqq \mathbf{u}_i$.

is completed. Instead, we replace node values of factual data with inferred values, $\hat{\mathbf{x}}$, by applying the currently learned set of structural equations $\hat{F}$ to the inferred exogenous node values, $\hat{\mathbf{u}}$. We use associational (*obs*), interventional (*do*), and counterfactual (*cf*) test datasets $\{\mathbf{X}^{obs}, \mathbf{X}^{do}, \mathbf{X}^{cf}\}$ to evaluate our method. The interventions required for $\mathbf{X}^{do}$ and $\mathbf{X}^{cf}$ are randomly sampled from $\boldsymbol{\mu}_i + \sigma(\mathbf{x}_i) \times \{-1.0, -0.5, -0.1, 0, 0.1, 0.5, 1.0\}$ to ensure realistic intervention values in the support of the each observed marginal distribution. Here, $\boldsymbol{\mu}_i$ and $\sigma(\mathbf{x}_i)$, represent the empirical mean and standard deviation of $\mathbf{x}_i$ from the training data. To generate an interventional or counterfactual sample, we perform do-operation on individual nodes only, one variable at a time.

For every SCM, we repeat experiments over five different seeds, each using a different PC model initialization. We report error metrics with mean and standard deviation, both multiplied by 100 for clarity. The PC graph is trained with 3000 samples for 1000 epochs with a batch size of 128. We use the vanilla stochastic gradient descent (*SGD*) optimizer for the node values with a learning rate of $\gamma = 3e - 3$ and $T = 8$ iterations for inference of node values during training and testing. For the weights, we use the *AdamW* optimizer with a learning rate of $\alpha = 8e - 3$ and a weight decay of $\lambda_w = 1e - 4$. For linear data, we fit the model using one-dimensional linear layers for each connection between the endogenous and exogenous variables of the SCM according to the causal structure defined given by the adjacency matrix $\mathbf{A}$. In the case of linear data, our approach does not require the use of neural networks with many hidden layers. This makes our causal PC graph transparent, efficient, and lightweight, because we only learn parameters that define the structural equations $F$ of the true data generating SCM. For the nonlinear experiments, we do not assume any detailed parametric knowledge about the SCMs and our method is feasible with general MLPs as structural equations. Hence, when learning nonlinear data, we replace the linear layers with small MLPs. Each MLP has 2 hidden layers with 16 neurons each and we use ELU (Clevert et al., 2015) as activation function. Note that, despite the large amount of epochs considered, every model converges in less than two minutes. All results are averaged over five random seeds.

**Metrics.** We now specify the metrics used to evaluate performance in estimating associational, interventional, and counterfactual distributions, as detailed in Section 3. Observational metrics are computed by comparing the true and estimated values of exogenous variables, using the available endogenous node data. Furthermore, note that for interventions and counterfactuals only the descendants, $des(\mathbf{x}_i)$, of an intervened node $\mathbf{x}_i$ are affected. Therefore, the causal order of a DAG becomes

important when assessing performance on interventional and counterfactual queries. As such, we report metrics with respect to the descendants of the intervention node, as per the adjacency matrix. We follow the works in the related literature (Sánchez-Martin et al., 2022; Chao et al., 2023) and report the following metrics:

- mean absolute error (MAE),
- maximum mean discrepancy (MMD) (Gretton et al., 2012),
- estimation squared error for the mean (MeanE),
- estimation squared error for standard deviation (StdE),
- mean of the squared error (MSE),
- standard deviation of the squared error (SSE).

We use MAE as a generic metric to assess the error between the estimated query and the ground truth query. The MMD metric is a sample-based distance measure between distributions. We use MMD to assess the match between the estimated distribution and the true distribution. The idea is to compare the means of both samples, $\widehat{\mathbf{X}}$ and $\mathbf{X}$, in a higher-dimensional feature space defined by a kernel function $k$.

For a pair of samples from each distribution, we compute the MMD as follows:

$$\text{MMD}(\mathbf{X}, \widehat{\mathbf{X}}) = \left\| \frac{1}{M} \sum_{i=1}^{M} \phi(\mathbf{x}^i) - \frac{1}{M} \sum_{i=1}^{M} \phi(\hat{\mathbf{x}}^i) \right\|^2 \tag{11}$$

$$= \frac{1}{M^2} \sum_{i=1}^{M} \sum_{j=1}^{M} k(\hat{\mathbf{x}}^i, \hat{\mathbf{x}}^j) - \frac{2}{M^2} \sum_{i=1}^{M} \sum_{j=1}^{M} k(\mathbf{x}^i, \hat{\mathbf{x}}^j) + \frac{1}{M^2} \sum_{i=1}^{M} \sum_{j=1}^{M} k(\mathbf{x}^i, \mathbf{x}^j). \tag{12}$$

Here, $\phi$ is the feature map of the kernel function $k$, and $\mathbf{x}^i$ and $\hat{\mathbf{x}}^i$ are the $i$-th samples from the ground truth and the inferred data, respectively. Each $\hat{\mathbf{x}}^i$ and $\mathbf{x}^i$ is a vector of $N$ features, one for each endogenous node in the DAG. The kernel function $k$ measures the similarity between data points in the feature space. In our implementation, we use a mixture of RBF (Gaussian) kernels with varying bandwidth parameters (Gretton et al., 2012).

We use MeanE and StdE to assess the estimated interventional distributions. MeanE and StdE measure the average squared error between the true and estimated mean and standard deviation of an interventional distribution, respectively. Both metrics are computed as averages across a set of intervention indices, $\mathcal{I}$, that correspond to nodes in the DAG that have descendants and thus are not leaf nodes.

Given the empirical means, $E[\mathbf{x}_i|do(\mathbf{x}_j)]$ and $E[\hat{\mathbf{x}}_i|do(\mathbf{x}_j)]$, and the empirical standard deviations, $SD[\mathbf{x}_i|do(\mathbf{x}_j)]$ and $SD[\hat{\mathbf{x}}_i|do(\mathbf{x}_j)]$, for node, $\mathbf{x}_i$, with intervention on node, $\mathbf{x}_j$, with index $j$, the MeanE and StdE are computed in the following way:

$$\text{MeanE} = \frac{1}{|\mathcal{I}|} \sum_{j \in \mathcal{I}} \frac{1}{|des(j)|} \sum_{i \in des(j)} \left( E[\mathbf{x}_i|do(\mathbf{x}_j)] - E[\hat{\mathbf{x}}_i|do(\mathbf{x}_j)] \right)^2, \tag{13}$$

$$\text{StdE} = \frac{1}{|\mathcal{I}|} \sum_{j \in \mathcal{I}} \frac{1}{|des(j)|} \sum_{i \in des(j)} \left( SD[\mathbf{x}_i|do(\mathbf{x}_j)] - SD[\hat{\mathbf{x}}_i|do(\mathbf{x}_j)] \right)^2. \tag{14}$$

We denote the number of intervention nodes in the DAG as $|\mathcal{I}|$, and $des(j)$ is the the set of descendants of the intervention node with index $j$. Finally, to assess the performance for the counterfactuals, we report the MSE and SSE for the descendants of an intervention node with index $j$. Both metrics are computed as averages across all intervention nodes in $\mathcal{I}$. We use the *Frobenius norm*, $T_j = \|\mathbf{x}_{des(j)} - \hat{\mathbf{x}}_{des(j)}\|_F$, to measure the difference between true and estimated values of a counterfactual query with intervention on node index $j$. Defining the average of the empirical mean

of $T_j$ as $E[T_j]$ and the average of the empirical standard deviation of $T_j$ as $SD[T_j]$, we retrieve the MSE and SSE metrics as:

$$\text{MSE} = \sum_{j \in \mathcal{I}} \frac{1}{|des(j)|} E[T_j], \tag{15}$$

$$\text{SSE} = \sum_{j \in \mathcal{I}} \frac{1}{|des(j)|} SD[T_j]. \tag{16}$$

To summarize, for associational inference, we report MMD on the observational test set. For interventional inference, we report MMD, MeanE, and StdE. For counterfactual inference, we report MSE and SSE. Additionally, we report MAE on the associational and interventional inference as well as MSE and SSE for our method's estimates of exogenous noise distributions, which are inferred in the abduction step while performing counterfactual inference. While not all the above metrics are required to evaluate a model's causal inference performance, we still include them for benchmark comparison against state-of-the-art methods (Sánchez-Martin et al., 2022; Khemakhem et al., 2021; Karimi et al., 2020). Across all metrics, lower values indicate better performance.

## D EXPERIMENTS ON COMMON CAUSAL GRAPHS

To generate data from a causal graph, we first sample a value for each of the $N$ exogenous variables that follow $\mathbf{u}_i \sim \mathcal{N}(\boldsymbol{\mu}_i, \sigma_i)$. Then, we use the deterministic structural equation, $F_i$, of node $\mathbf{x}_i$ to compute its value as $x_i := F_i(par(x_i), \mathbf{u}_i)$. Each $F_i$ is a linear equation with additive noise of the form $F_i = \sum_{j \in par(\mathbf{x}_i)} w_{ji} \mathbf{x}_j + \mathbf{u}_i$, where $par(\mathbf{x}_i)$ denotes the direct parents of node $\mathbf{x}_i$ according to the causal graph structures provided in Fig. 7. We follow the same procedure for the non-linear SCM, however, instead of using linear structural equations, $F_i$, we use non-linear structural equations with additive noise. The non-linear structural equations used to generate the non-linear SCM data are shown in Table 1.

| Graph | $F_1 := X_1$ | $F_2 := X_2$ | $F_3 := X_3$ | $F_4 := X_4$ | $F_5 := X_5$ |
|---|---|---|---|---|---|
| Fork | $U_1$ | $-1 + \frac{3}{1+\exp(-2X_1)} + U_2$ | $0.25X_1^2 + U_3$ | - | - |
| Collider | $U_1$ | $U_2$ | $0.05X_1 + 0.25X_2^2 + U_3$ | - | - |
| Confounder | $U_1$ | $-1 + \frac{3}{1+\exp(-2X_1)} + U_2$ | $X_1 + 0.25X_2^2 + U3$ | - | - |
| Chain | $U_1$ | $-1 + \frac{3}{1+\exp(-2X_1)} + U_2$ | $0.25X_2^2 + U3$ | - | - |
| Mediator | $U_1$ | $1 - \cosh(0.5X_1) + U_2$ | $X_1 + 0.25X_2^2 + U3$ | - | - |
| M-bias | $U_1$ | $U_2$ | $0.5X_1^2 - X_2 + U3$ | $X_1 + 0.5X_1^2 + U4$ | $-1.5X_2^2 + U_5$ |
| Butterfly | $U_1$ | $U_2$ | $0.5X_1^2 - X_2 + U3$ | $X_1 + 0.5X_1^2 - 0.25X_3^2 + U4$ | $-1.5X_2^2 + 0.25X_3^2 + U_5$ |

Table 1: Structural equations for non-linear SCM data generation.

For the causal inference experiments with the common causal graphs in Fig. 7 as well as the butterfly graph presented in the main body of the paper, we focused on performing interventions on nodes that are interesting. By interesting we mean that we want to show experimental results for interventions and counterfactuals on nodes that are neither root nodes nor leaf nodes. The reason being that interventions on such nodes either correspond to (a) regular conditional (associational) queries, as is the case with interventions on root nodes or (b) counterfactual queries that are not differentiable from interventional queries, as is the case for interventions on leaf nodes. Consequently, we provide results for the following causal inference scenarios: (i) chain graph with intervention on the node $\mathbf{x}_2$, (ii) confounder graph with intervention on node $\mathbf{x}_2$, (iii) collider graph with intervention on root node $\mathbf{x}_1$, (iv) fork graph with intervention on node $\mathbf{x}_1$, (v) mediator graph with intervention on node $\mathbf{x}_2$, (vi) M-bias graph with intervention on node $\mathbf{x}_1$, and (vii) butterfly bias graph with intervention on node $\mathbf{x}_3$. The $\mathbf{x}_3$ intervention in the butterfly graph is interesting and challenging because $\mathbf{x}_3$ is a collider and confounder at the same time. Finally, to provide a better understanding of the datasets, what an intervention entails, and how the associational, interventional, and counterfactual distributions differ from each other for each of the causal graphs depicted in Fig. 7, we provide the

histograms of each *graph-intervention* scenario in Figs. 11 to 14. The distribution of each exogenous variable is depicted in the first row as $\mathbf{u}_i$. The histograms show the difference between observational distribution (second row), interventional distribution (third row), and counterfactual distribution (last row), which are denoted as $\mathbf{x}^{obs}$, $\mathbf{x}_i^{do(\mathbf{x}_j)}$, $\mathbf{x}_i^{\mathbf{x}_i'}$, respectively (for the purpose of these figures).

**Results.** The experiments in this section display that our method is able to infer correctly associational, interventional, and counterfactual distributions. More specifically, we show how we can (1) learn the parameters of the SCM (structural equations and exogenous distributions) and (2) deploy the error nodes of a fitted PC model in such a way that allows us to manipulate a structural equation to answer causal queries. First, in Figs. 15 and 16, we show the convergence of our predictive coding network while learning the parameters of the SCM for all three and five node causal graphs. In the left column, we can see that our method converges for all graph structures and that we do not overfit the training data. Moreover, we observed that a low free energy does not correspond to a converged model. The MAE continues to decrease, while the energy changes minimally after 50 epochs. The right column shows that the convergence is stable and smooth among all nodes in the causal graph. We show the free energy by node for all exogenous and endogenous variables. Second, in Figs. 17 and 18, we show the causal inference performance of our method by tracking the MAE for associational, interventional, and counterfactual test queries throughout the SCM learning process. We perform interventions on all types of nodes to show that our model is able to correctly infer causal queries on: root nodes with no parents, intermediate nodes with parents and children, and leaf nodes with no children. Note that the left column represents the associational inference error on the exogenous nodes only because during training we are provided with data of the endogenous variables. Third, in Figs. 19 and 20, we plot the MAE metric for test interventions and counterfactuals during the SCM learning process. We choose intervention nodes that are non-trivial by selecting, wherever available, intervention nodes that are neither root nor leaf nodes.

To conclude, in Table 2, we summarize our results and compare our method to state-of-the-art models on all common causal graphs of this work. First, we see that our method consistently outperforms all state-of-the-art methods, on all causal graph structures, for all types of causal queries. Second, we see that a further advantage of our method is that our PC network is parameter efficient as it requires only a fraction of the parameters required for the state-of-the-art methods (Sánchez-Martin et al., 2022; Khemakhem et al., 2021; Karimi et al., 2020), which employ thousands of parameters to infer causal distributions. For example, in the linear case, we only need to learn the parameters of the exogenous distributions plus one parameter for each adjacency weight that connects two nodes in the causal graphs.

**Discussion.** Our proposed method does not require more parameters than the number of parameters that define the true structural equations of the SCM. As such, our model is lightweight and simple to train. Having explored various hyperparameters, we found that our model is not prone to overfitting nor does it require hyperparameter tuning or model selection to infer causal distributions. The metrics reported in Table 2 show that our method is consistent with the increasing complexity of DAG structures and able to well capture observational, interventional and counterfactual distributions for all graphs. We experimented with varying Gaussian distribution parameters for the exogenous variables and found that our causal inference approach is robust to arbitrary Gaussians. As such, our method does not require the assumption of standard normally distributed exogenous variables as is the case in some of the related literature (Sánchez-Martin et al., 2022; Saha & Garain, 2022; Chao et al., 2023). Furthermore, our model is minimal as in being the simplest model that adequately explains the data which is shown by the number of parameters our model architectures require. Therefore, we provide an Occam's razor like solution (Blumer et al., 1987) for causal inference with linear SCMs. Finally, we do not rely on complex approximators, such as GNN, VAE, gradient boosted regressor or normalizing flow models that require extensive hyperparameter tuning, to learn causal relationships between the observed variables.

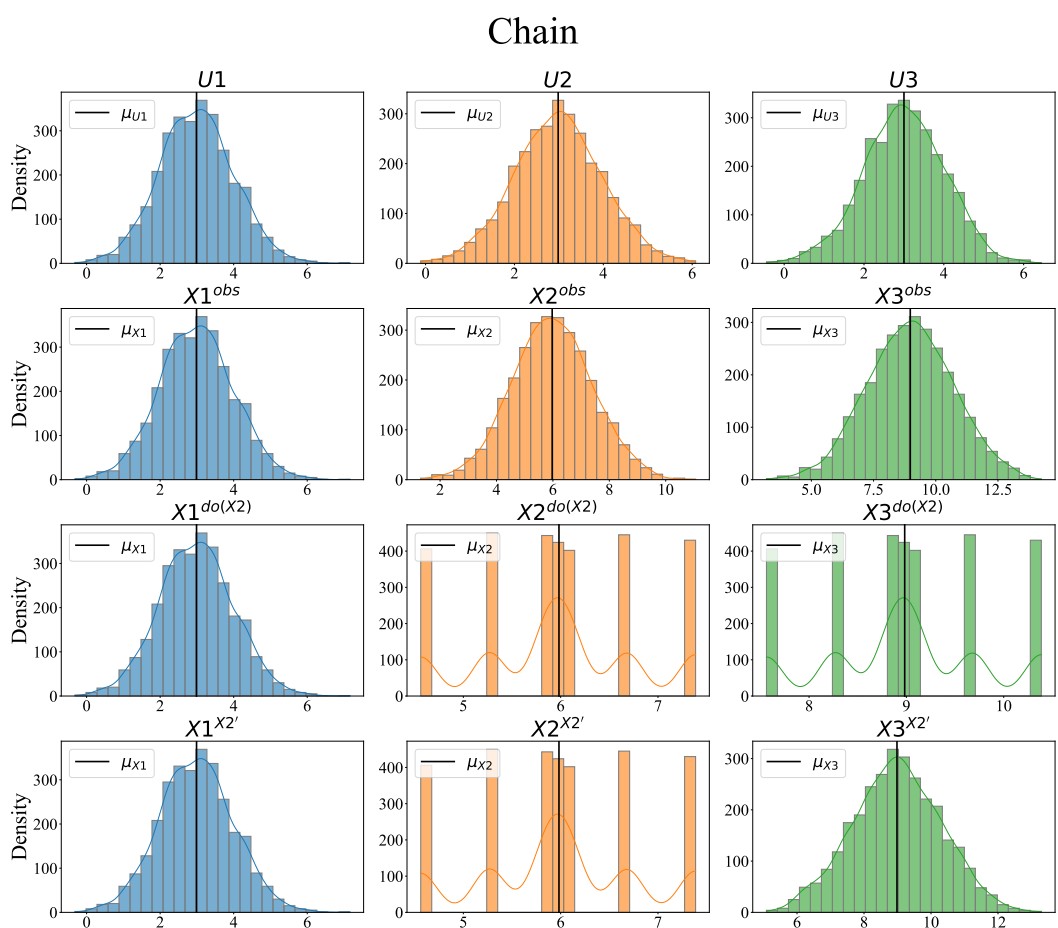

Figure 8: Causal hierarchy of distributions for chain SCM with intervention on $\mathbf{x}_2$. First row: Exogenous distribution. Second row: Associational distribution. Third row: Interventional distribution. Last row: Counterfactual distribution.

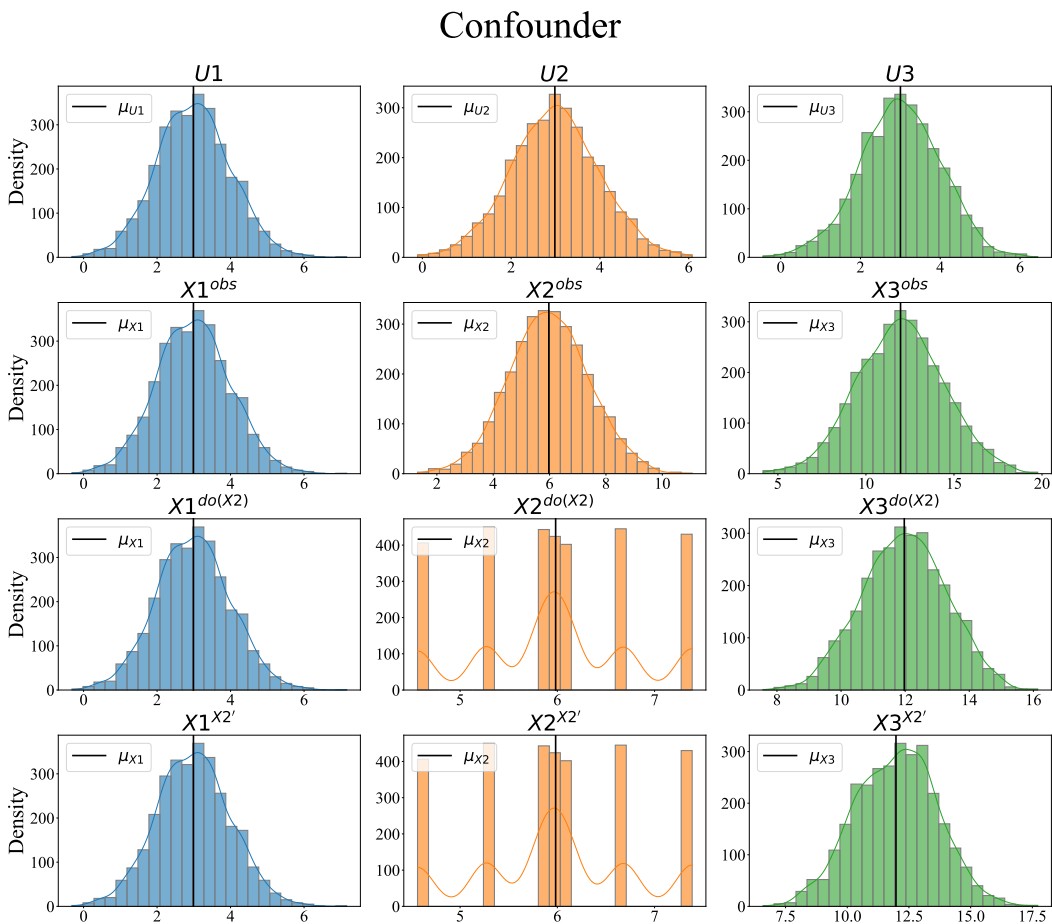

Figure 9: Causal hierarchy of distributions for confounder SCM with intervention on $\mathbf{x}_2$. First row: Exogenous distribution. Second row: Associational distribution. Third row: Interventional distribution. Last row: Counterfactual distribution.

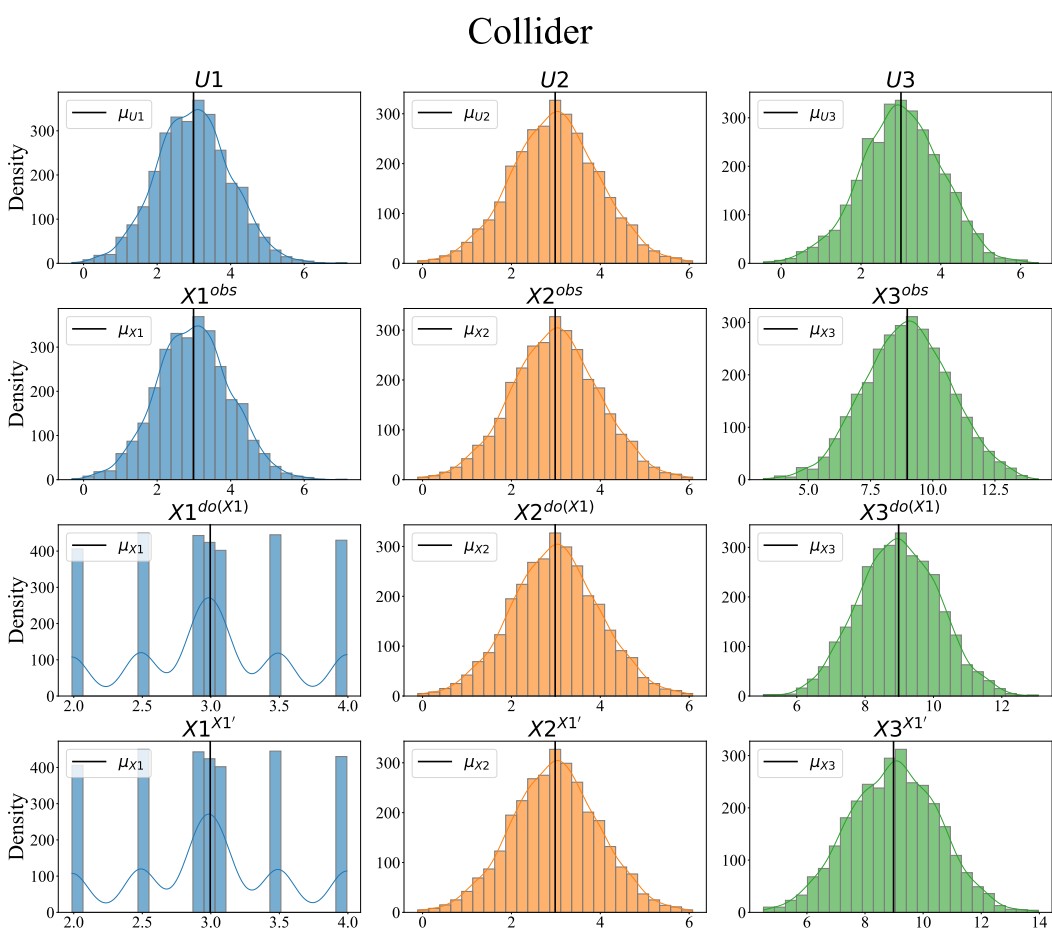

Figure 10: Causal hierarchy of distributions for collider SCM with intervention on $\mathbf{x}_1$. First row: Exogenous distribution. Second row: Associational distribution. Third row: Interventional distribution. Last row: Counterfactual distribution.

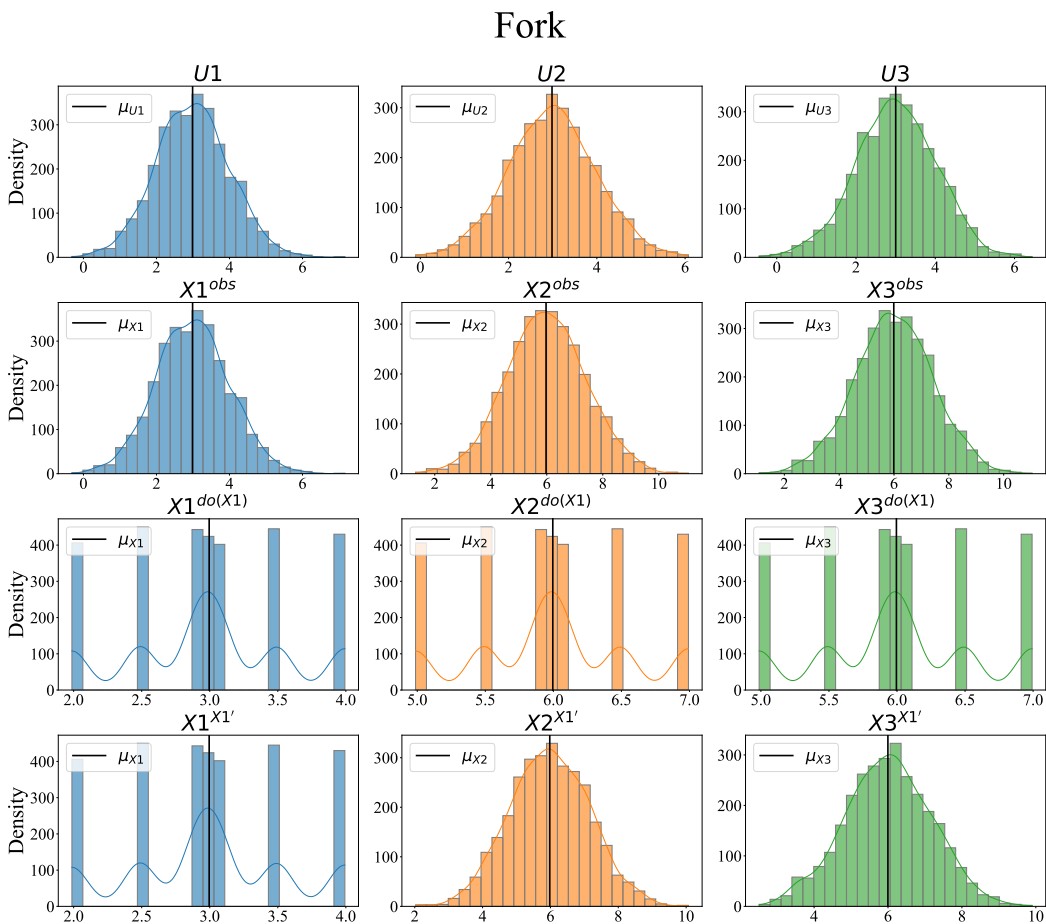

Figure 11: Causal hierarchy of distributions for fork SCM with intervention on $\mathbf{x}_1$. First row: Exogenous distribution. Second row: Associational distribution. Third row: Interventional distribution. Last row: Counterfactual distribution.

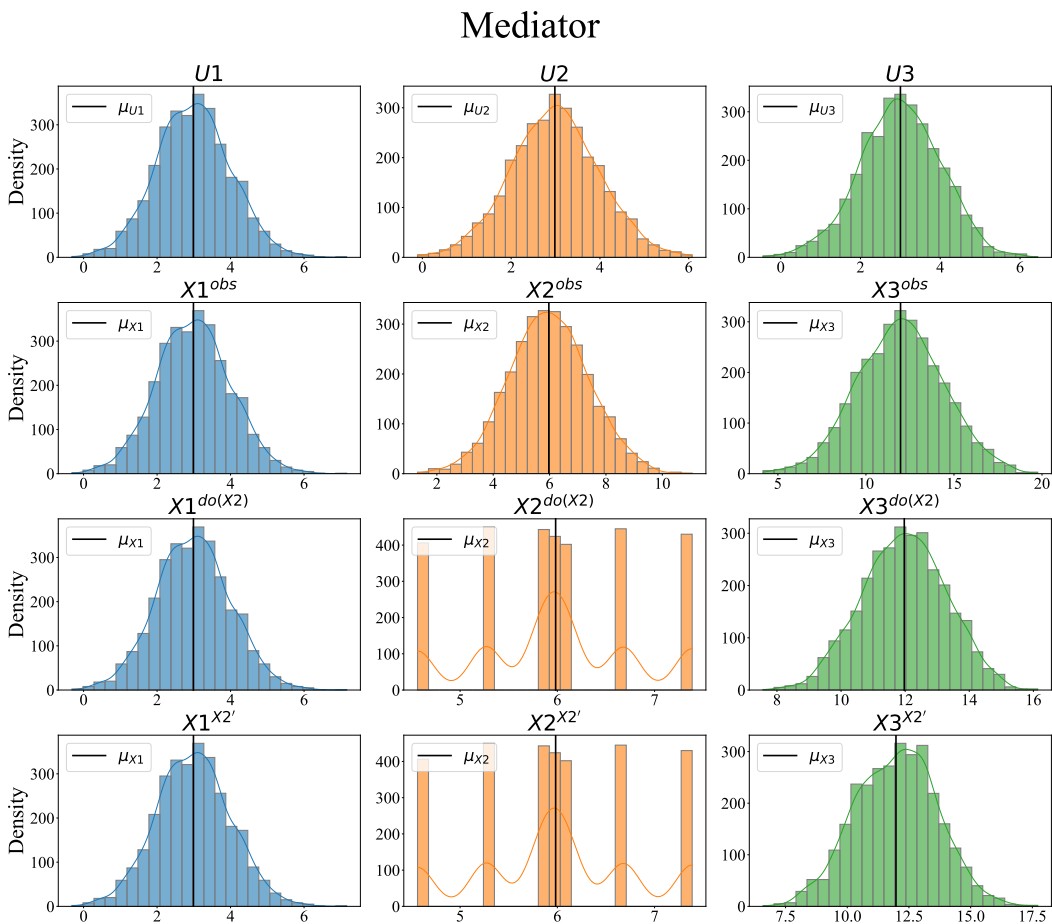

Figure 12: Causal hierarchy of distributions for mediator SCM with intervention on $\mathbf{x}_2$. First row: Exogenous distribution. Second row: Associational distribution. Third row: Interventional distribution. Last row: Counterfactual distribution.

M

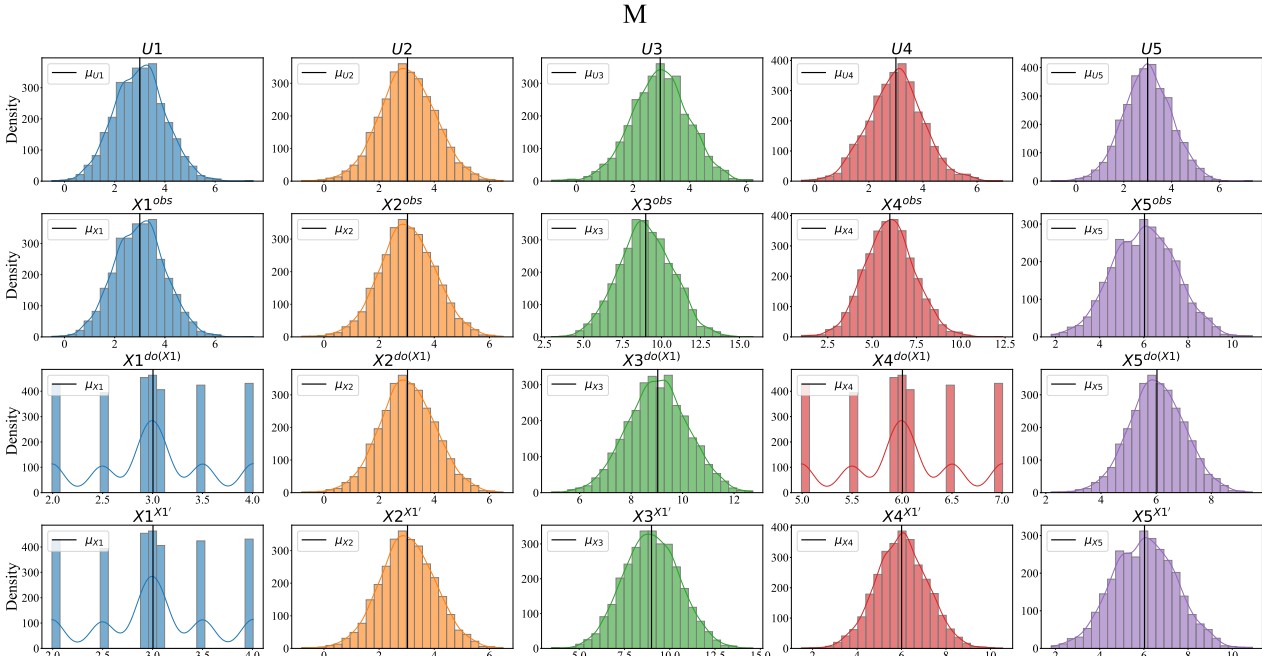

Figure 13: Causal hierarchy of distributions for M-bias SCM with intervention on $\mathbf{x}_1$. First row: Exogenous distribution. Second row: Associational distribution. Third row: Interventional distribution. Last row: Counterfactual distribution.

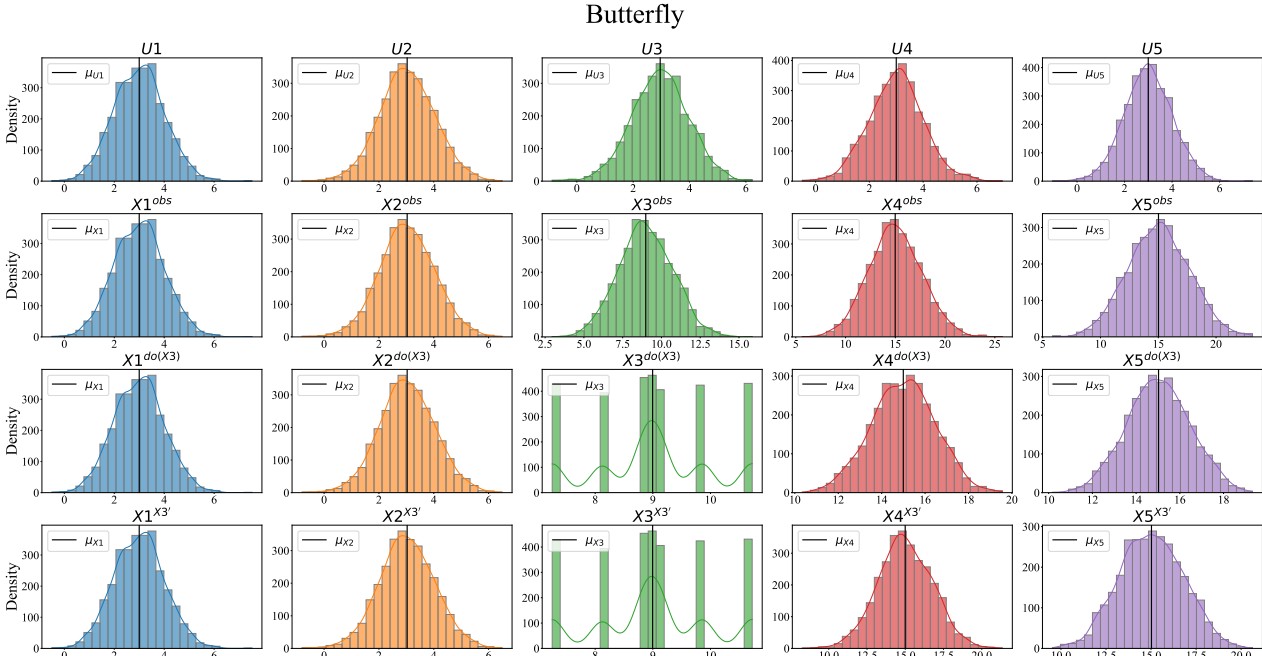

Figure 14: Causal hierarchy of distributions for butterfly SCM with intervention on $\mathbf{x}_3$. First row: Exogenous distribution. Second row: Associational distribution. Third row: Interventional distribution. Last row: Counterfactual distribution.

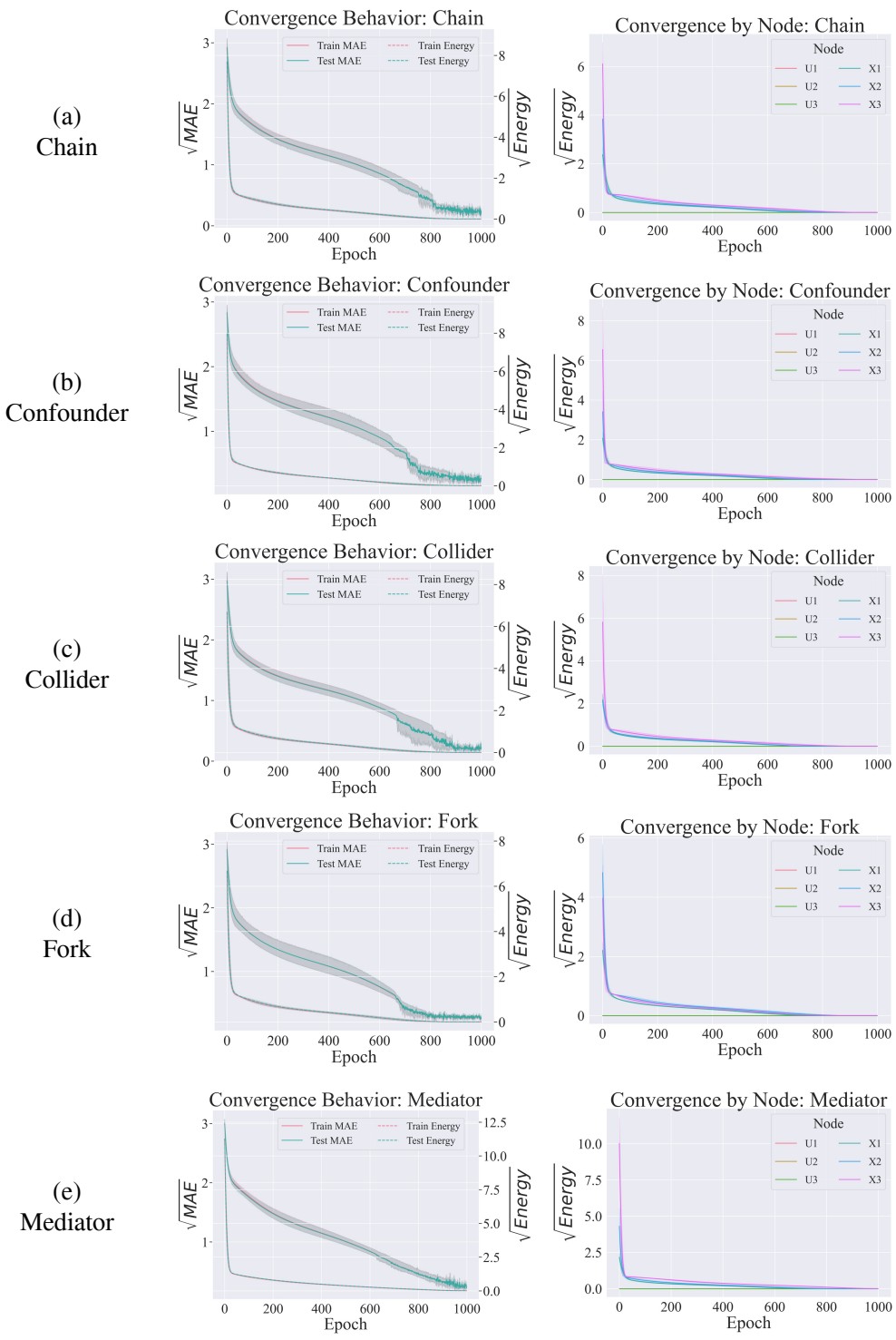

Figure 15: Convergence of energy and MAE by node for causal graphs with three nodes. Left column: Convergence of total train and test MAE in comparison to free energy. Right column: Energy by node.

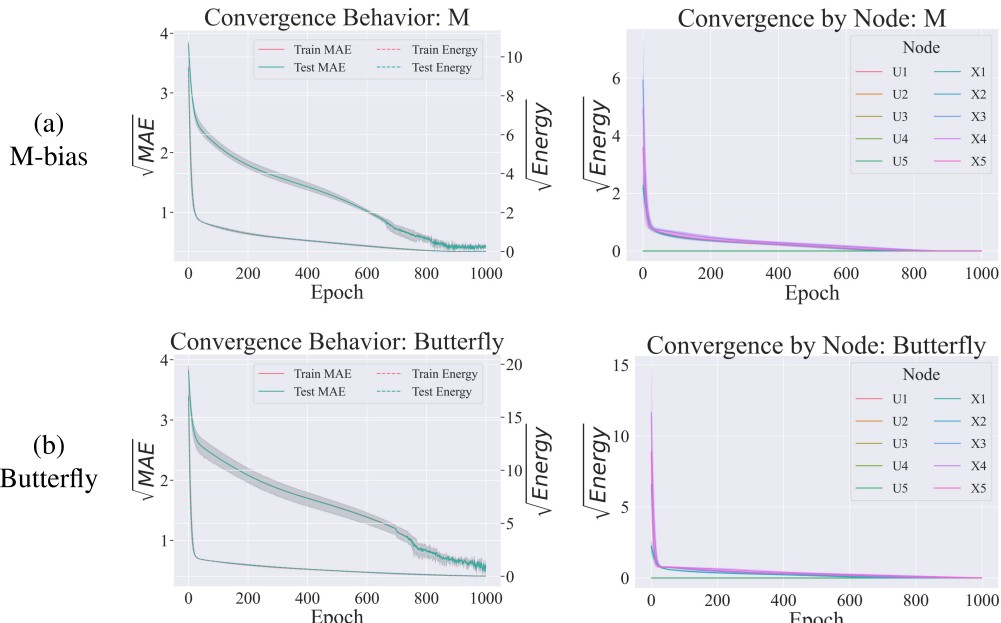

Figure 16: Convergence of energy and MAE by node for causal graphs with five nodes. Left column: Convergence of total train and test MAE in comparison to free energy. Right column: Energy by node.

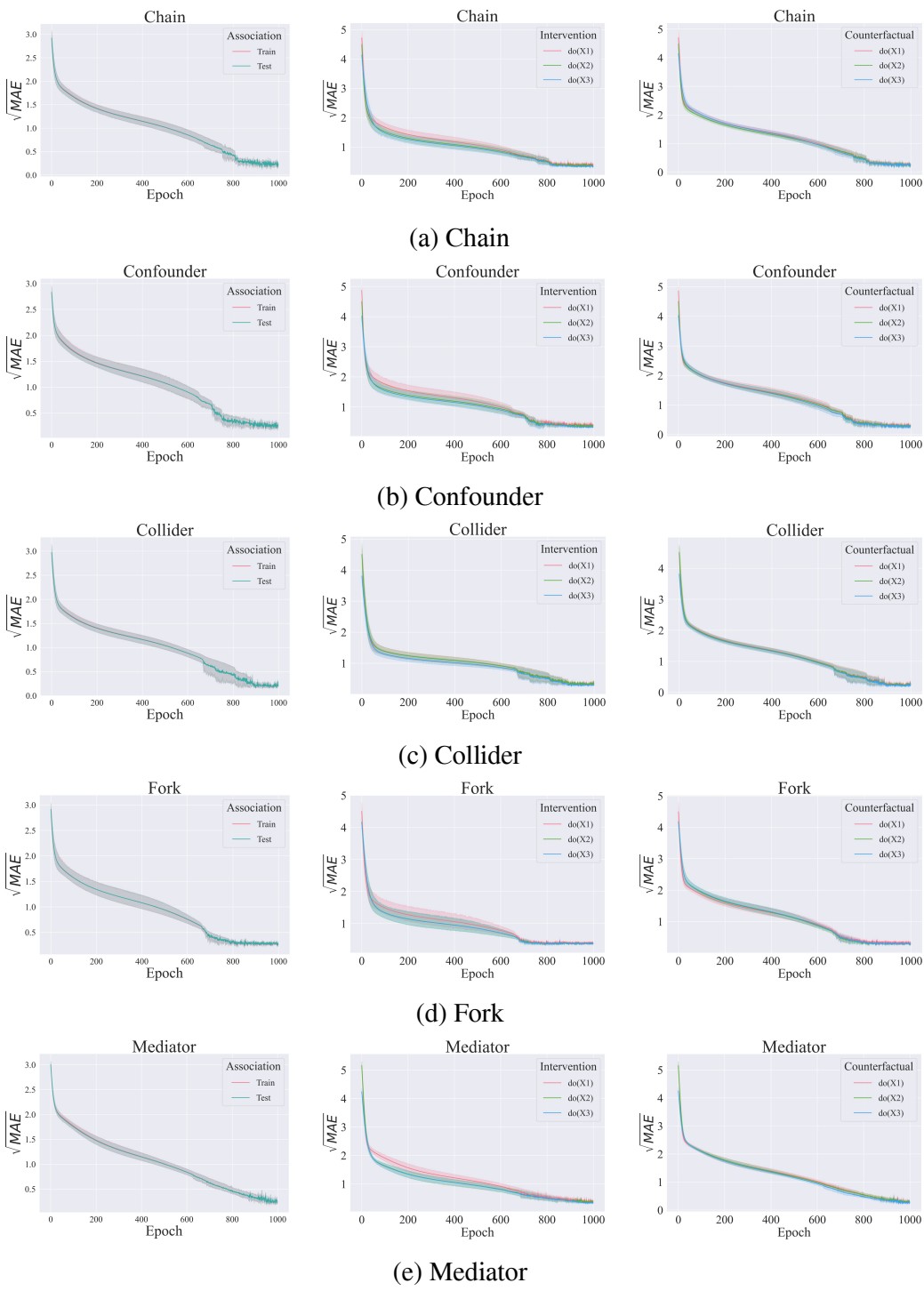

Figure 17: Causal inference performance throughout SCM learning process. We track MAE for all interventions of all three node causal graphs. For association plots (left column) inference is performed on exogenous nodes given factual test data.

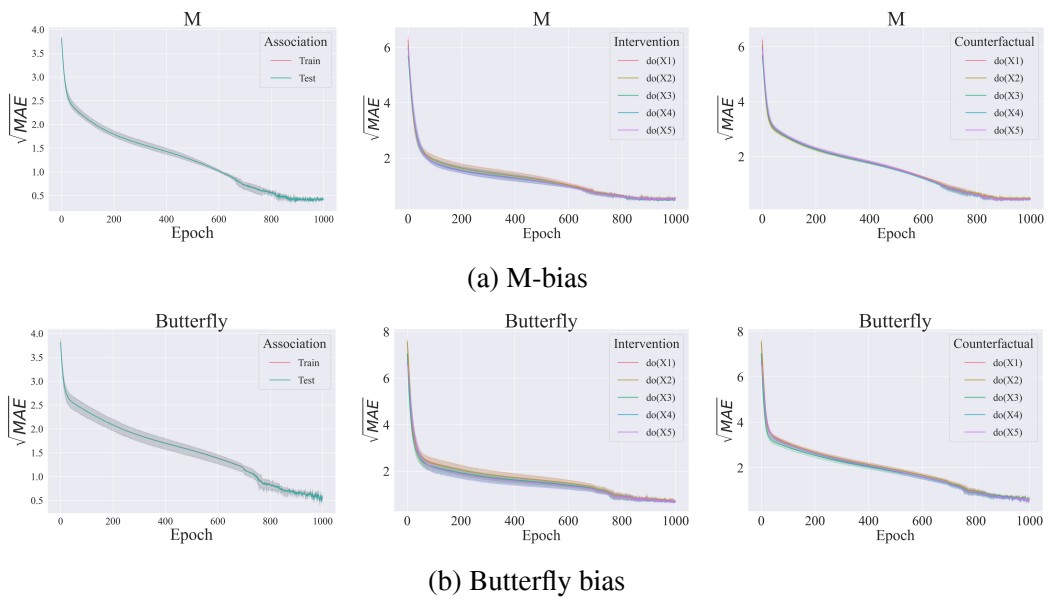

(a) M-bias

(b) Butterfly bias

Figure 18: Causal inference performance throughout SCM learning process. We track MAE for all interventions of all five node causal graphs. For association plots (left column) inference is performed on exogenous nodes given factual test data.

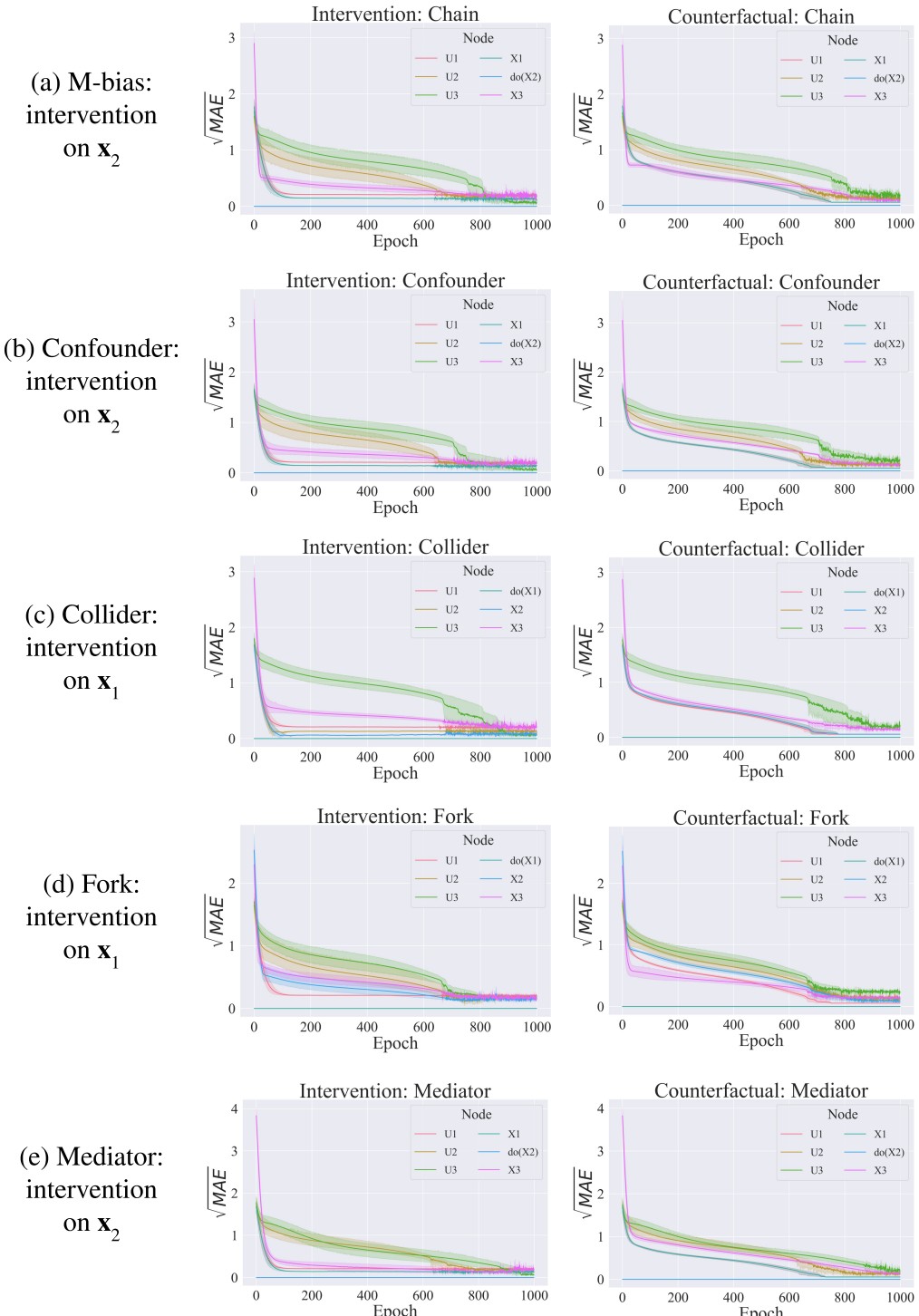

Figure 19: Performance of interventional and counterfactual inference throughout SCM learning process. For each three node causal graph we choose a specific intervention node, if available a node that is neither a root nor leaf node, and track MAE by node.

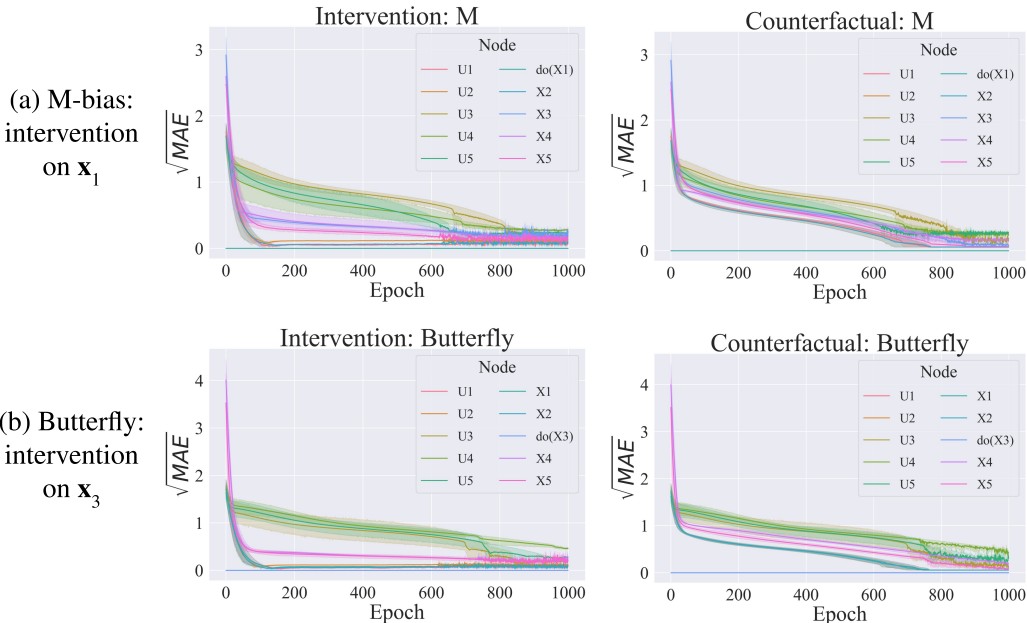

Figure 20: Performance of interventional and counterfactual inference throughout SCM learning process. For each five node causal graph we choose a specific intervention node, if available a node that is neither a root nor leaf node, and track MAE by node.

| SCM | Model | Observational MMD ↓ | Interventional MMD ↓ | MeanE ↓ | StdE ↓ | Counterfactual MSE ↓ | SSE ↓ | Params. # ↓ |
|---|---|---|---|---|---|---|---|---|
| chain (LIN) | Ours | **0.02 ± 0.02** | **0.09 ± 0.11** | 0.25 ± 0.30 | 0.00 ± 0.00 | **0.97 ± 0.52** | 0.68 ± 0.36 | **8** |
| | MultiCVAE | 7.98 ± 1.16 | 62.66 ± 6.47 | 75.74 ± 3.94 | 40.53 ± 1.09 | 57.74 ± 12.24 | 25.92 ± 7.01 | 7145 |
| | CAREFL | 14.81 ± 0.63 | 19.70 ± 0.28 | 1.29 ± 0.40 | 89.53 ± 1.88 | 36.58 ± 5.17 | 29.78 ± 4.10 | 1488 |
| | VACA | 4.73 ± 1.01 | 14.65 ± 4.54 | 8.34 ± 2.43 | 19.78 ± 0.72 | 98.21 ± 8.85 | 6.08 ± 2.06 | 2045 |
| chain (NLIN) | Ours | **0.35 ± 0.03** | **0.34 ± 0.30** | 0.26 ± 0.22 | 0.02 ± 0.01 | **5.81 ± 1.36** | 6.90 ± 0.76 | **648** |
| | MultiCVAE | 9.25 ± 3.33 | 86.64 ± 6.13 | 129.40 ± 8.63 | 86.03 ± 1.65 | 22.66 ± 5.05 | 12.39 ± 2.50 | 7145 |
| | CAREFL | 15.56 ± 4.35 | 12.01 ± 1.89 | 1.00 ± 0.26 | 84.36 ± 4.43 | 34.12 ± 9.98 | 27.53 ± 6.93 | 1488 |
| | VACA | 1.97 ± 0.82 | 8.86 ± 5.90 | 0.76 ± 0.39 | 18.03 ± 0.56 | 8.24 ± 1.94 | 8.04 ± 0.44 | 2045 |
| confounder (LIN) | Ours | **0.07 ± 0.08** | **0.23 ± 0.32** | 1.07 ± 1.35 | 0.02 ± 0.02 | **2.18 ± 1.00** | 1.57 ± 0.73 | **9** |
| | MultiCVAE | 4.35 ± 0.63 | 56.68 ± 3.86 | 135.05 ± 5.03 | 45.62 ± 1.00 | 52.40 ± 11.85 | 23.11 ± 3.65 | 7209 |
| | CAREFL | 16.48 ± 0.72 | 26.04 ± 1.03 | 0.91 ± 0.38 | 96.90 ± 2.56 | 32.59 ± 6.41 | 32.05 ± 10.56 | 1488 |
| | VACA | 4.48 ± 1.59 | 10.94 ± 6.06 | 5.43 ± 1.56 | 17.85 ± 0.49 | 78.80 ± 14.23 | 5.34 ± 1.58 | 2454 |
| confounder (NLIN) | Ours | **0.34 ± 0.11** | **0.09 ± 0.05** | 0.17 ± 0.12 | 0.03 ± 0.01 | **4.64 ± 1.02** | 5.65 ± 0.85 | **969** |
| | MultiCVAE | 7.70 ± 0.55 | 61.17 ± 3.60 | 146.56 ± 11.79 | 76.75 ± 1.38 | 29.15 ± 4.46 | 16.64 ± 2.05 | 7209 |
| | CAREFL | 13.90 ± 2.48 | 12.94 ± 2.11 | 1.02 ± 0.17 | 84.52 ± 4.19 | 32.94 ± 12.42 | 25.37 ± 8.61 | 1488 |
| | VACA | 4.43 ± 3.72 | 8.75 ± 11.57 | 1.75 ± 2.55 | 19.18 ± 1.57 | 19.27 ± 2.89 | 8.70 ± 1.01 | 2454 |
| collider (LIN) | Ours | **0.01 ± 0.00** | **0.04 ± 0.03** | 0.11 ± 0.08 | 0.01 ± 0.00 | **1.48 ± 0.08** | 1.10 ± 0.06 | **8** |
| | MultiCVAE | 10.41 ± 1.25 | 47.12 ± 10.20 | 58.73 ± 3.57 | 68.06 ± 3.22 | 81.63 ± 15.99 | 49.35 ± 7.12 | 7145 |
| | CAREFL | 12.39 ± 0.73 | 10.50 ± 0.14 | 1.15 ± 0.31 | 94.69 ± 3.53 | 31.06 ± 5.18 | 30.51 ± 5.13 | 1488 |
| | VACA | 4.13 ± 2.70 | 9.79 ± 6.99 | 4.96 ± 3.02 | 34.51 ± 0.78 | 97.74 ± 21.20 | 12.53 ± 2.82 | 2045 |
| collider (NLIN) | Ours | **0.33 ± 0.06** | **0.02 ± 0.01** | 0.03 ± 0.02 | 0.01 ± 0.01 | **2.05 ± 0.28** | 2.86 ± 0.29 | **648** |
| | MultiCVAE | 14.01 ± 5.91 | 62.31 ± 10.55 | 70.09 ± 16.79 | 78.85 ± 3.06 | 45.48 ± 5.72 | 33.83 ± 4.77 | 7145 |
| | CAREFL | 13.38 ± 2.37 | 9.06 ± 1.22 | 0.64 ± 0.16 | 96.07 ± 4.46 | 29.72 ± 7.02 | 29.87 ± 8.78 | 1488 |
| | VACA | 9.15 ± 10.17 | 11.32 ± 4.45 | 2.69 ± 1.06 | 34.03 ± 0.68 | 12.67 ± 1.90 | 8.07 ± 1.10 | 2045 |
| fork (LIN) | Ours | **0.03 ± 0.01** | **0.04 ± 0.03** | 0.06 ± 0.06 | 0.01 ± 0.00 | **2.02 ± 0.41** | 1.41 ± 0.32 | **8** |
| | MultiCVAE | 11.28 ± 1.44 | 46.91 ± 10.52 | 59.04 ± 4.35 | 67.76 ± 3.32 | 81.16 ± 15.48 | 49.10 ± 7.04 | 7145 |
| | CAREFL | 11.39 ± 0.65 | 10.44 ± 0.47 | 0.77 ± 0.25 | 94.62 ± 3.68 | 29.51 ± 5.34 | 31.22 ± 2.88 | 1488 |
| | VACA | 5.26 ± 3.12 | 9.19 ± 7.00 | 4.67 ± 2.92 | 34.41 ± 0.72 | 97.62 ± 23.12 | 13.97 ± 3.10 | 2045 |
| fork (NLIN) | Ours | **0.23 ± 0.03** | **0.04 ± 0.03** | 0.01 ± 0.01 | 0.03 ± 0.02 | **4.56 ± 1.12** | 5.00 ± 1.18 | **648** |
| | MultiCVAE | 9.57 ± 2.55 | 57.24 ± 7.56 | 62.72 ± 4.39 | 18.06 ± 0.98 | 143.31 ± 25.70 | 64.05 ± 12.13 | 7145 |
| | CAREFL | 11.72 ± 2.31 | 12.57 ± 2.07 | 0.93 ± 0.29 | 66.72 ± 2.58 | 88.89 ± 3.00 | 70.55 ± 1.70 | 1488 |
| | VACA | 5.22 ± 1.18 | 6.85 ± 2.79 | 2.00 ± 0.72 | 13.73 ± 0.25 | 103.97 ± 14.61 | 39.92 ± 6.15 | 2045 |
| mediator (LIN) | Ours | **0.02 ± 0.01** | **0.08 ± 0.08** | 0.28 ± 0.28 | 0.01 ± 0.00 | **1.59 ± 0.24** | 1.11 ± 0.18 | **9** |
| | MultiCVAE | 5.93 ± 0.88 | 55.31 ± 3.57 | 134.42 ± 5.23 | 46.64 ± 1.50 | 52.18 ± 11.70 | 23.19 ± 3.89 | 7209 |
| | CAREFL | 13.60 ± 0.64 | 26.04 ± 1.03 | 0.91 ± 0.38 | 96.90 ± 2.56 | 34.93 ± 8.19 | 36.05 ± 13.23 | 1488 |
| | VACA | 6.25 ± 2.06 | 10.94 ± 6.06 | 5.43 ± 1.56 | 17.85 ± 0.49 | 78.69 ± 14.21 | 5.61 ± 1.70 | 2454 |
| mediator (NLIN) | Ours | **0.27 ± 0.06** | **0.05 ± 0.04** | 0.04 ± 0.05 | 0.03 ± 0.03 | **5.55 ± 0.56** | 6.62 ± 0.95 | **969** |
| | MultiCVAE | 10.39 ± 3.34 | 56.63 ± 13.31 | 46.48 ± 2.68 | 53.77 ± 6.83 | 1187.64 ± 109.46 | 2044.01 ± 334.22 | 7209 |
| | CAREFL | 17.10 ± 4.38 | 13.22 ± 1.30 | 3.21 ± 0.53 | 89.96 ± 3.33 | 1001.70 ± 2.95 | 1753.10 ± 2.59 | 1488 |
| | VACA | 4.90 ± 2.92 | 16.87 ± 3.64 | 6.38 ± 1.07 | 21.08 ± 0.74 | 1072.91 ± 13.33 | 1732.35 ± 2.07 | 2454 |
| M-bias (LIN) | Ours | **0.02 ± 0.00** | **0.07 ± 0.07** | 0.09 ± 0.12 | 0.01 ± 0.00 | **1.75 ± 0.12** | 1.16 ± 0.09 | **14** |
| | MultiCVAE | 16.20 ± 2.69 | 63.95 ± 7.35 | 58.02 ± 1.90 | 32.36 ± 2.73 | 47.15 ± 10.02 | 19.55 ± 4.99 | 11951 |
| | CAREFL | 19.47 ± 1.63 | 15.62 ± 0.74 | 0.92 ± 0.23 | 67.21 ± 3.04 | 26.39 ± 2.63 | 20.73 ± 0.58 | 3080 |
| | VACA | 2.50 ± 0.46 | 6.60 ± 1.53 | 3.35 ± 0.93 | 12.53 ± 0.22 | 62.26 ± 4.97 | 6.53 ± 2.36 | 3681 |
| M-bias (NLIN) | Ours | **0.31 ± 0.01** | **0.19 ± 0.04** | 0.31 ± 0.09 | 0.06 ± 0.04 | **7.83 ± 0.71** | 6.27 ± 0.44 | **1294** |
| | MultiCVAE | 20.06 ± 6.48 | 397.97 ± 12.76 | 4197.87 ± 98.51 | 205.02 ± 36.38 | 268.85 ± 5.36 | 64.28 ± 7.94 | 11951 |
| | CAREFL | 21.09 ± 3.33 | 258.57 ± 2.50 | 189.33 ± 5.01 | 105.67 ± 2.86 | 240.42 ± 1.67 | 80.93 ± 0.58 | 3080 |
| | VACA | 3.01 ± 0.65 | 384.20 ± 7.41 | 191.00 ± 3.45 | 13.97 ± 0.59 | 234.23 ± 11.24 | 52.50 ± 3.51 | 3681 |
| butterfly (LIN) | Ours | **0.14 ± 0.02** | **0.21 ± 0.14** | 0.41 ± 0.25 | 0.03 ± 0.00 | **4.66 ± 1.33** | 3.24 ± 1.02 | **16** |
| | MultiCVAE | 16.85 ± 3.31 | 83.44 ± 10.15 | 139.98 ± 5.83 | 79.49 ± 11.86 | 55.45 ± 3.50 | 24.49 ± 2.64 | 12079 |
| | CAREFL | 22.03 ± 2.28 | 38.93 ± 1.16 | 1.23 ± 0.17 | 88.59 ± 3.31 | 23.52 ± 3.20 | 18.14 ± 1.48 | 3080 |
| | VACA | 4.16 ± 0.55 | 6.89 ± 1.25 | 3.83 ± 0.90 | 8.73 ± 0.11 | 57.40 ± 3.84 | 4.16 ± 0.94 | 4499 |
| butterfly (NLIN) | Ours | **0.32 ± 0.01** | **0.73 ± 0.24** | 0.67 ± 0.25 | 0.58 ± 0.42 | **19.77 ± 1.43** | 17.46 ± 1.55 | **1936** |
| | MultiCVAE | 17.28 ± 4.59 | 119.05 ± 16.82 | 2067.99 ± 86.70 | 2107.67 ± 398.55 | 355.31 ± 7.86 | 262.08 ± 2.56 | 12079 |
| | CAREFL | 24.79 ± 3.97 | 30.66 ± 2.30 | 2.68 ± 0.39 | 85.71 ± 3.33 | 297.32 ± 3.19 | 267.70 ± 1.06 | 3080 |
| | VACA | 4.01 ± 0.41 | 13.47 ± 3.77 | 3.65 ± 1.07 | 12.11 ± 1.24 | 385.53 ± 17.39 | 235.63 ± 3.87 | 4499 |

Table 2: Comparing our model with state-of-the-art methods across various SCM structures for observational, interventional, and counterfactual distributions. All values are scaled by 100. Mean and standard deviation are calculated over five seeds. LIN and NLIN denote linear and nonlinear SCMs, respectively.

# E  CLASSIFICATION EXPERIMENTS

Here, we provide all the information needed to reproduce the results of the classification experiments provided in the main body of the paper. Furthermore, we also provide a more detailed study on how the results are affected when changing the parameters of the model.

**Setup.**  The primary focus of our experiments is on the training of fully-connected neural network models with 2000 neurons on two datasets: MNIST and FashionMNIST. The chosen architecture for the models is a simple fully connected PC graph. For the models, we conducted an exhaustive hyperparameter search. We opted for a grid search approach, examining several combinations of learning rates for the weights and the latent variables, and subsequently training the models for each combination. The chosen learning rates for the weights were $\{0.0001, 0.00005, 0.00001\}$. As for the latent variables, the learning rates tested were $\{1, 0.5\}$, and every batch of 128 examples was observed for $T \in \{3, 5, 7\}$ iterations. To optimize the weights of our models, we have used the Adam optimizer; for the value nodes, we have used SGD; as an activation function, ReLU. To conclude, have also tested *incremental predictive coding* (iPC), a variation of PC that updates the weight parameters at every time step $t$. This method has been shown to improve both the performance and the stability of predictive coding models (Salvatori et al., 2022c). Training was performed for 20 epochs for each combination of learning rates in the grid search. It is important to note that training always converged before the 20th epoch, ensuring a stable model for each hyperparameter combination. At every epoch of the training process, we have computed the test accuracy using both interventional queries and conditional queries.

**Results.**  The results of our experiments showed a clear pattern: regardless of the combination of hyperparameters, interventional queries consistently outperformed conditional queries. This pattern was observed across all models and datasets, suggesting that interventional queries might be a more effective tool for PC graphs. The best results were obtained using a learning rate of the value nodes of 0.5, and $T = 3$. The learning rate of the parameters slightly affected the performance, unless we consider values outside the proposed range. As a learning algorithm, we observe that iPC is indeed more performing and stable, as shown in Figure 21.

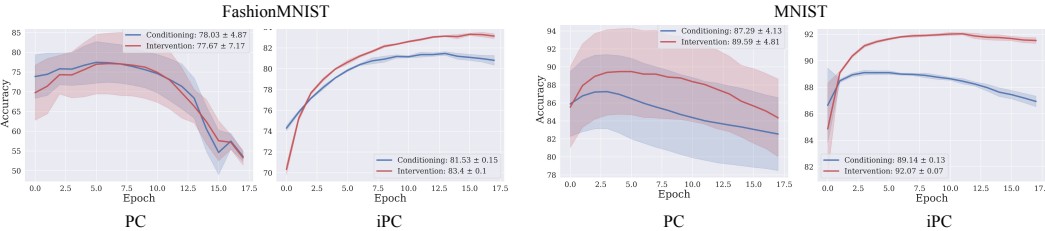

Figure 21: Difference in performance and stability of iPC and PC on classification tasks.

# F    ROBUSTNESS EXPERIMENTS

Recent work has shown that available deep-learning-based methods fail to obtain sufficient performance on counterfactual queries under specific circumstances, and proposed novel techniques to overcome this shortcoming (De Brouwer, 2022). Here, we show that predictive coding achieves the same state-of-the-art results, while requiring a simpler architecture with no ad hoc training techniques. To do so, we test PC graphs on the colored-MNIST dataset introduced in (De Brouwer, 2022). The dataset consists of tuples $(\mathbf{x}, \mathbf{u}_z, \mathcal{T}, \mathbf{y}, \mathcal{T}', \mathbf{y}')$, where $\mathbf{x}$ is the original MNIST image, $\mathcal{T}$ is the assigned treatment, $\mathbf{u}_z$ is a hidden exogenous random variable that determines the color of the observed outcome image $\mathbf{y}$, and $\mathbf{y}'$ is the counterfactual response obtained when applying the alternative treatment $\mathcal{T}'$.

**Setup.**    To replicate the experiment, we have used a PC graph with a structure that is equivalent to the 4-nodes SCM used in the original work, and trained it with Algorithm 1. We used nodes with the ground truth number of dimensions (i.e., 784, 1, and 784 respectively) to represent the observed variables $\mathbf{x}$, $\mathcal{T}$, and $\mathbf{y}$. Instead, we left the dimension of the remaining hidden node $h$ as a hyperparameter $d_h$ as the value $\mathbf{u}_z$ is never observed by the model. The edges between the nodes in the PC graph represent feedforward networks. Figure 5 summarizes the architecture used. We experimented with different depths, widths, and activation functions, without experiencing any unexpected results (e.g., deeper and wider networks would have slightly better performance). The architecture can be seen as an encoder-decoder structure. The nodes $\mathbf{x}$, $\mathcal{T}$, and $\mathbf{u}_z$ are encoded using respectively 3, 1, and 1 fully connected layers with a hidden dimensions of 1024 and *tanh* as activation function. Then, the embedding is decoded into $\mathbf{y}$ using 3 other fully connected layers with the same hidden dimension of 1024. The experiment was conducted as follows:

- During training, we fix the nodes $\mathbf{x}$, $\mathcal{T}$, and $\mathbf{y}$ to the corresponding observed variables and we initialize $h$ to 0. We train for 128 epochs with a batch size of 256. We train using *iPC* and $T = 16$. We set the nodes learning rate to $\gamma = 0.005$ and the weights learning rate to $\alpha = 0.00005$. We use the *SGD* optimizer for the nodes and the *Adamw* optimizer for the weights. Consequently, the model has never direct access to any of $\mathbf{u}_z$, $\mathcal{T}'$, or $\mathbf{y}'$.

- The inference process is divided in two phases. Firstly, we repeat the same procedure as above, while setting $\alpha = 0.0$, so that the weights of the model are not changed. This allows the network to adapt to the provided $\mathbf{y}$ by storing its extra information (i.e., the color, in this instance) in the hidden node $\mathbf{u}_z$. Secondly, for each sample in a batch, after $T = 16$ steps, we replace $\mathcal{T}$ with $\mathcal{T}'$ to compute the counterfactual $\mathbf{y}'$ and compare it with the ground truth image. To obtain $\mathbf{y}'$, we simply forward through the network the information stored in the nodes $\mathbf{x}$, $\mathcal{T}$, and $\mathbf{u}_z$ during the first phase. To produce the digits in Fig. 5, we fixed the node $\mathcal{T}$ to each angle $\in \{0°, 10°, 20°, 30°, 40°\}$. Furthermore, our model is able to produce counterfactual not only by modifying the rotation $\mathcal{T}$, but also the color encoded by $\mathbf{u}_z$. To show this, we take the value of the node $\mathbf{u}_z$ computed for the a sample and use it to generate all the remaining $\mathbf{y}'$ in the batch.

**Results.**    We obtain results comparable with the ones in (De Brouwer, 2022). Our method has the advantage of using a straightforward multi-layer perceptron architecture trained with an unmodified version of the predictive coding learning algorithm. This shows the capability and versatility of predictive coding to work in various tasks in which other deep-learning techniques tend to fail, such as Diff-SCM (Sanchez & Tsaftaris, 2022), Deep-SCM (Pawlowski et al., 2020), and Deep-ITE (Shalit et al., 2017). Figure 5 in the main body shows a magnified example of counterfactual reconstructions that demonstrate that our method is robust with respect to interventions on either rotation or color. Compared to (De Brouwer, 2022), we are able to generalize to rotations of $40°$, even if this introduces some noise in the generated output. Furthermore, contrary to the model presented in (De Brouwer, 2022), our architecture is robust with respect to the choice of the hyperparameter linked to $u_z$ and does not necessitate to perform a hyperparameter sweep to find the right value.

# G  STRUCTURE LEARNING

## G.1  EXPERIMENTS ON RANDOM GRAPHS

In this section, we show results on the convergence behavior of structure learning with a PC graph to understand the relationship between variational free energy and the approximation error of the weighted adjacency matrix. Furthermore, we describe in detail the metrics used to evaluate the estimated weighted adjacency matrix as well as accuracy metrics for assessing the learned relationships and directions of the adjacency matrix. We also provide all details on the model and training parameters used to reproduce our structure learning experiments. Finally, we compare our method against established structure learning algorithms for random graphs of various types and complexities.

**Setup.**  Our causal structure learning method only requires observational data as input. The two types of random graphs that we consider for our experiments are (1) Erdős-Rényi (ER) with either 1 or 2 expected edges per node, denoted as ER1 and ER2, and (2) scale-free (SF) graphs with either 2 or 4 expected edges per node, denoted as SF2 and SF4, respectively. We use graphs with $N \in \{10, 15, 20\}$ nodes and generate datasets with 2000 samples. These two graph types are selected to demonstrate the versatility and robustness of our method in handling various graph structures.

We generate synthetic data by first sampling a binary adjacency matrix, $\mathbf{A}$, for a DAG. Next, we place uniformly random edge weights onto the binary adjacency matrix, to obtain a weighted adjacency matrix, $\mathbf{W}$. Finally, we sample observational data based on a set of linear structural equations with additive Gaussian noise, $\mathbf{u} \sim \mathcal{N}(\mathbf{0}, \mathbf{I}_N)$, such that

$$\mathbf{x} = \mathbf{W}^T \mathbf{x} + \mathbf{u} \in \mathbb{R}^N.$$

To fit the PC model to the observational data, we use the stochastic gradient descent (*SGD*) optimizer for the node values with a learning rate of $\gamma = 1e-4$ and $T = 16$. For the weights, we use the *Adamw* optimizer with a learning rate of $\alpha = 5e-3$. We enforce two penalties onto our learning algorithm. First, a DAG penalty to ensure that the discovered graph is acyclic and directed, as proposed in (Zheng et al., 2018). Second, an L1 penalty that encourages the PC network to find a causal structure that is sparse. We add both penalties into the predictive coding objective. The penalties are each weighted by $\lambda_{L1} = 5e-6$ and $\lambda_{DAG} = 200$, for L1 and DAG penalty, respectively.

**Structure learning metrics**  Here, we describe the metrics used to evaluate the performance for estimating causal graph structures in the experiments of Section 4. First, for the weighted adjacency matrix of a DAG with $N$ nodes, we report the mean absolute error (MAE) between the true, $\mathbf{W}$, and the estimated, $\widehat{\mathbf{W}}$, weighted adjacency matrix as the average of the absolute differences between corresponding entries in the two matrices:

$$\text{MAE}(\mathbf{W}, \widehat{\mathbf{W}}) = \frac{1}{N^2} \sum_{i=1}^{N} \sum_{j=1}^{N} |\mathbf{W}_{ij} - \widehat{\mathbf{W}}_{ij}|. \tag{17}$$

Second, to evaluate the correctness of the learned edge directions and the adjacency relationships in the causal graph, we report metrics on the estimated binary adjacency matrix, $\widehat{\mathbf{A}}$, that is obtained via thresholding as follows:

$$\widehat{\mathbf{A}}_{ij} = \begin{cases} 1 & \text{if } \widehat{\mathbf{W}}_{ij} > \omega \\ 0 & \text{otherwise.} \end{cases}$$

We use the same data as proposed in (Zheng et al., 2018), and we follow their procedure and use $\omega = 0.3$ in all experiments. We report the following graph metrics: (i) F-score (F1), (ii) structural Hamming distance (SHD), (iii) false discovery rate (FDR), (iv) true positive rate (TPR), (iv) false positive rate (FPR), and (v) number of directed edges discovered (NNZ). Each metric is computed between the true adjacency matrix, $\mathbf{A}$, and the estimated adjacency matrix, $\widehat{\mathbf{A}}$. To compute each metric, we first need the following quantities:

- true positive (TP): a discovered edge, with correct direction,
- reverse (R): a discovered edge, with incorrect direction,

- false positive (FP): a discovered edge, not present in $\mathbf{A}$,
- true negative (TN): a non-discovered edge, not present in $\mathbf{A}$,
- false negative (FN): a non-discovered edge, present in $\mathbf{A}$,
- missing (M): a non-discovered edge, present in $\mathbf{A}$.

Based on these quantities, each metric is computed as follows:

- F1 $= \frac{2\text{TP}}{2\text{TP}+\text{FP}+\text{FN}}$,
- SHD $= \text{R} + \text{M} + \text{FP} = \sum_{i=1}^{N} \sum_{j=1}^{N} |\mathbf{A}_{ij} - \widehat{\mathbf{A}}_{ij}|$,
- FDR $= \frac{\text{FP}+\text{R}}{\text{FP}+\text{TP}}$,
- FPR $= \frac{\text{FP}+\text{R}}{\text{FP}+\text{TN}}$,
- TPR $= \frac{\text{TP}}{\text{TP}+\text{FN}}$,
- NNZ $= \text{TP} + \text{FP}$.

**Results.** First, we show results on learning the causal structure for the two most difficult graphs of our experiments, namely, ER2 and SF4 graphs, each with $N = 20$. We see that PC graphs are able to learn good approximations of the ground truth weighted adjacency matrix of complex random graphs. This is depicted in the left columns of Figs. 22 and 23, respectively. The right columns in Figs. 22 and 23 shows how the MAE decreases as the predictive coding objective converges. In all cases, we observe that while the energy converges early on, the causal discovery performance (MAE) keeps improving. Second, in Table 3, we compare our method against established and recent causal discovery algorithms. In contrast to the baselines, our method consistently exhibits a good performance across various graph structures and does not deteriorate strongly for graphs of varying complexity by maintaining its causal structure learning abilities despite irregular node degree distributions and an increasing number of edges and nodes. This is reflected in the high accuracy metrics (F1) and low structural hamming distance (SHD) obtained with our method. From our experiments, we observed that our method performs very well on scale-free graphs different to most of the benchmarks, which struggle on such random graphs. To conclude, the causal structure learning experiments conducted demonstrate that our method can learn arbitrary DAG structures of varying characteristics and levels of complexity. The complexity is determined by factors such as the number of nodes in the graph and the degree distribution of each node. Furthermore, our method demonstrates a robust performance even in the face of challenges such as increasingly uneven degree distributions and growing node cardinality. This distinguishes our approach from the baseline methods, thereby further highlighting the effectiveness of our predictive coding framework for causal structure learning.

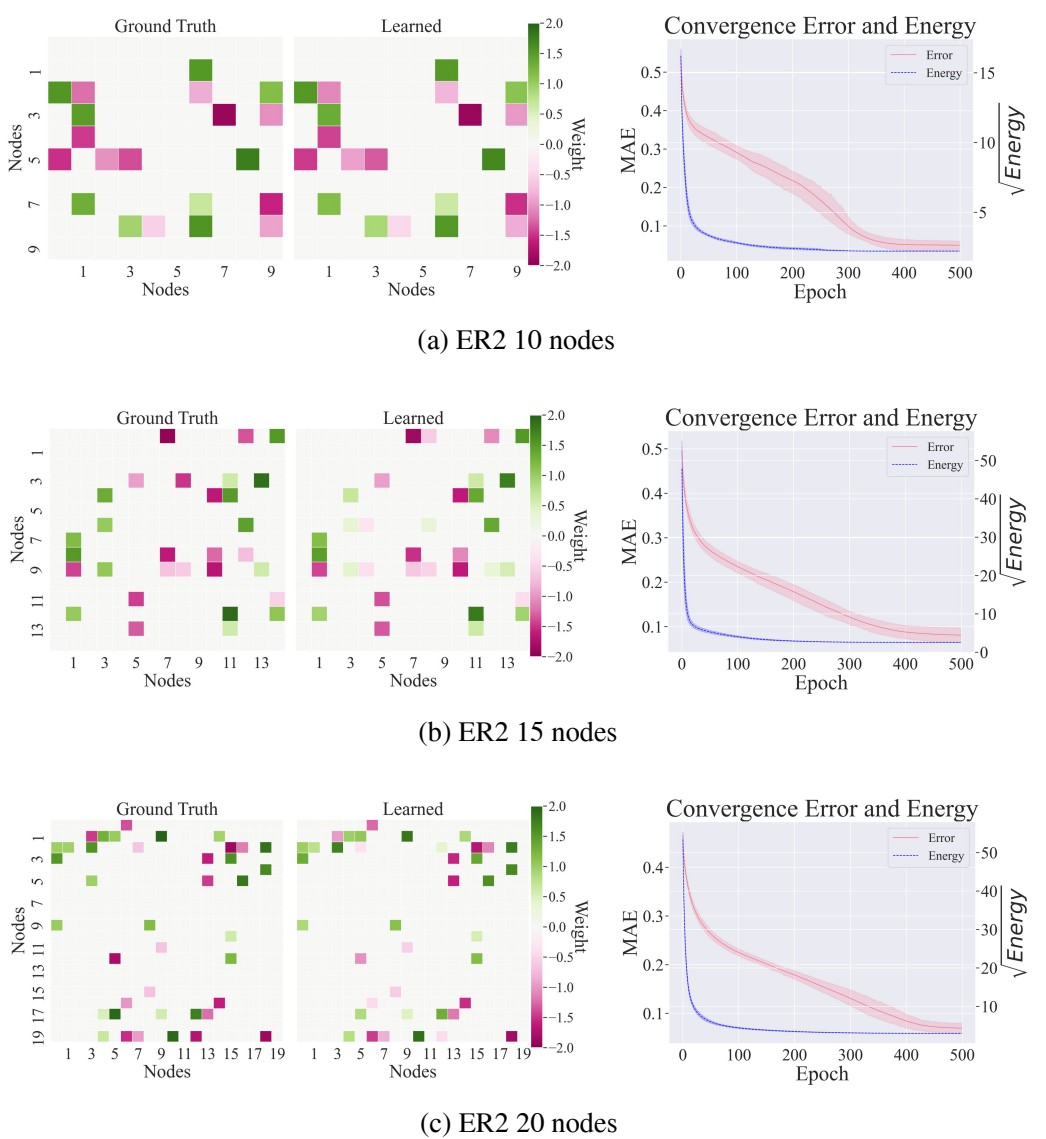

Figure 22: Learned structures and convergence behavior (energy vs. MAE) for ER2 graphs of various complexity.

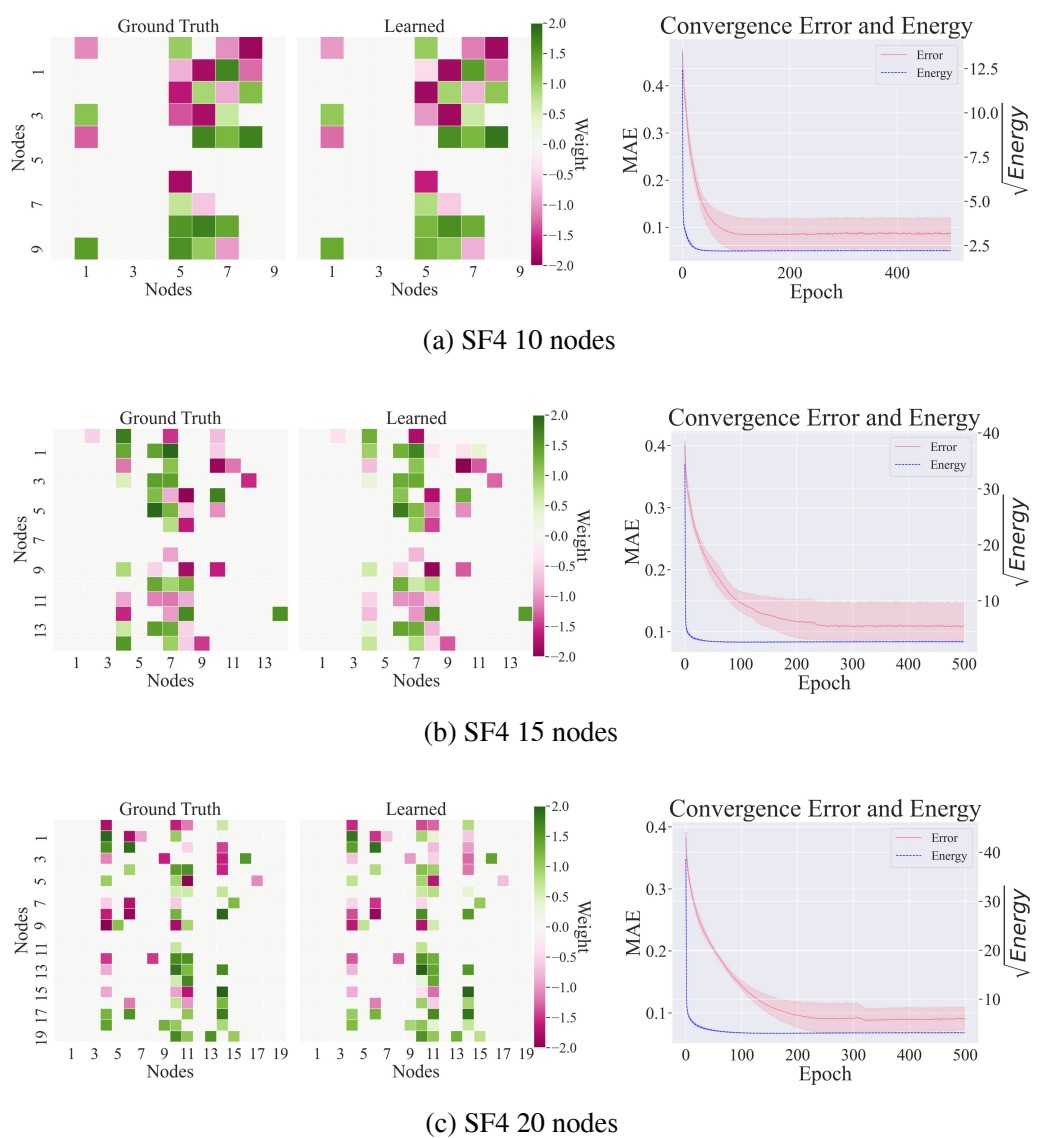

(a) SF4 10 nodes

(b) SF4 15 nodes

(c) SF4 20 nodes

Figure 23: Learned structures and convergence behavior (energy vs. MAE) for SF4 graphs of various complexity.

### G.2 CLASSIFICATION

In the main body of this work, we showed that pruning unnecessary connections in a complete graph results in a hierarchical structure with improved performance. In this section, we provide further details to reproduce our results.

The classification experiments are performed on the MNIST, FashionMNIST, and 2-MNIST datasets. The latter is obtained by pairing the image $\hat{\mathbf{x}}$ in each sample $(\hat{\mathbf{x}}, \hat{\mathbf{y}})$ of the MNIST dataset with a new digit image $\hat{\mathbf{x}}'$ sampled uniformly from the dataset (while maintaining the train and test splitting intact). Thus, a data point of the 2-MNIST dataset consists of the tuple $(\hat{\mathbf{x}}, \hat{\mathbf{x}}', \hat{\mathbf{y}})$.

We consider a predictive coding graph with 6 nodes, one of dimension 784 (input dimension), one of dimension 10 (output dimension), and 4 hidden nodes of dimension $d$. In the case of 2-MNIST, we have two nodes of dimension 784, and only three of dimension $d$. The results reported in this work were obtained with $d = 128$. During training, we used a batch size of 512 and $T = 32$. To start from a complete graph, as the one defined in Section 3, we define a fully connected layer $f_{i,j}$, with $gelu$ activation function, going from node $i$ to node $j$ for each ordered pair of nodes $(i, j)$. The output of each layer $f_{i,j}$ is then multiplied by a scaling factor $a_{i,j}$ that determines the strength of the connection from node $i$ to node $j$. Together, the factors $a_{i,j}$ determine the adjacency matrix $A$. To enforce sparse connectivity and prune unnecessary edges, we add to the matrix $A$ the L1 regularizer $l(A)$. Consequently, the loss function to optimize becomes $\mathcal{L} = F + \omega \cdot l(A)$, where $\omega$ is a weighting factor.

**Degenerate Example.** We start our discussion by showing a degenerate example, which arises when we do not use either negative examples, or a prior that forces an acyclic structure, but only the prior $l(A)$ which enforces sparsity. In this case, the modes is unable to learn the causal dependency between input and output, and converges towards a degenerate structure, where each output node predicts itself via a cyclic structure, which can be a self loop, or a closed loop with length larger than one. As each node either predicts itself or is unused, the total variational free energy of the network is going to be close to 0, despite the network being randomly guessing the output. An example of such structures is provided in Fig. 6. This shows the importance of additional methods, that force the network to be aware of the causal dependency between the input and the output. We now test the two proposed methods: the acyclic prior, and the use of negative examples.

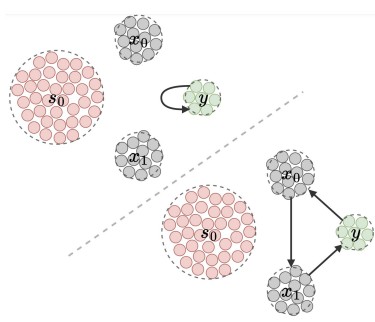

Figure 24: Examples of degenerate networks, where the label predicts itself either via self loops, or via cycles.

To overcome this, we propose two different methods:

1. *force an acyclic structure*, by adding to the loss function the regulariser $h(A) = tr(exp(A \times A))$ introduced in (Zheng et al., 2018). We weight $h(A)$ by a scalar $\eta$.

2. *force a connection between input nodes and output nodes*, by introducing negative samples in the training dataset: with probability $p_{ns}$ we sample randomly a new label $\hat{y}_{ns}$. We modify the energy function:

$$\hat{F} = \sum_{i \neq i_{\mathbf{y}}} \|\mathbf{x}_i - \boldsymbol{\mu}_i\|^2 + (\|\mathbf{x}_{i_y} - \mu_{i_y}\|^2 - k)^2, \tag{18}$$

where $i_y$ is the index of the node fixed to $\hat{\mathbf{y}}$ and $k$ the new energy target. We set $k = 0$ for positive samples and $k > 0$ for negative samples. With negative samples, the output node cannot simply learn to predict itself, as the energy would be non-zero for negative samples, for which the energy target is $k > 0$.

**Discussion.** Both methods produce hierarchical structures that achieve a better performance than the original complete graph. Method (1) has the disadvantage of introducing an inductive bias in the architecture by completely removing loops and requiring a complex balance between the $\omega$ and $\eta$ parameters, as they affect each other. On the other hand, method (2), despite overcoming these issues, seems more brittle with respect of the choice of hyperparameters and produces a smaller variety of networks. The value of $\omega$ determines the overall network structure. For method (1),

| model | $N$ | graph | FDR ↓ | TPR ↑ | FPR ↓ | SHD ↑ | NNZ - | F1 ↑ |
|---|---|---|---|---|---|---|---|---|
| Ours | | | 0.08 ± 0.04 | 0.92 ± 0.04 | 0.02 ± 0.01 | **0.80 ± 0.45** | 10.00 ± 0.00 | **0.92 ± 0.04** |
| GES | | | 0.22 ± 0.08 | 0.88 ± 0.11 | 0.07 ± 0.03 | 3.00 ± 1.41 | 11.20 ± 0.45 | 0.68 ± 0.09 |
| PC | 10 | ER1 | 0.13 ± 0.04 | 0.82 ± 0.04 | 0.03 ± 0.01 | 2.20 ± 0.45 | 9.40 ± 0.55 | 0.77 ± 0.03 |
| ICALiNGAM | | | 0.25 ± 0.15 | 0.86 ± 0.11 | 0.09 ± 0.05 | 3.20 ± 2.05 | 11.60 ± 1.14 | 0.80 ± 0.13 |
| NOTEARS | | | 0.08 ± 0.08 | 0.90 ± 0.07 | 0.02 ± 0.02 | 1.20 ± 0.84 | 9.80 ± 0.84 | 0.91 ± 0.07 |
| Ours | | | 0.02 ± 0.04 | 0.94 ± 0.04 | 0.02 ± 0.04 | **1.40 ± 0.89** | 19.20 ± 1.30 | **0.96 ± 0.03** |
| GES | | | 0.86 ± 0.03 | 0.26 ± 0.05 | 1.30 ± 0.02 | 33.60 ± 0.55 | 37.60 ± 1.14 | 0.17 ± 0.03 |
| PC | 10 | ER2 | 0.47 ± 0.09 | 0.45 ± 0.08 | 0.32 ± 0.06 | 13.20 ± 1.48 | 17.00 ± 0.71 | 0.47 ± 0.08 |
| ICALiNGAM | | | 0.31 ± 0.14 | 0.74 ± 0.12 | 0.27 ± 0.13 | 9.20 ± 4.76 | 21.60 ± 1.14 | 0.71 ± 0.13 |
| NOTEARS | | | 0.14 ± 0.00 | 0.90 ± 0.00 | 0.12 ± 0.00 | 4.00 ± 0.00 | 21.00 ± 0.00 | 0.88 ± 0.00 |
| Ours | | | 0.03 ± 0.04 | 0.99 ± 0.03 | 0.00 ± 0.01 | **0.60 ± 0.89** | 15.20 ± 0.45 | **0.98 ± 0.03** |
| GES | | | 0.22 ± 0.14 | 0.83 ± 0.10 | 0.04 ± 0.03 | 5.00 ± 3.00 | 16.00 ± 1.00 | 0.80 ± 0.12 |
| PC | 15 | ER1 | 0.16 ± 0.05 | 0.85 ± 0.03 | 0.03 ± 0.01 | 3.40 ± 0.89 | 15.20 ± 0.45 | 0.78 ± 0.03 |
| ICALiNGAM | | | 0.30 ± 0.06 | 0.77 ± 0.09 | 0.06 ± 0.01 | 5.40 ± 1.14 | 16.60 ± 1.52 | 0.73 ± 0.07 |
| NOTEARS | | | 0.06 ± 0.08 | 0.89 ± 0.10 | 0.01 ± 0.01 | 2.00 ± 2.00 | 14.20 ± 0.84 | 0.92 ± 0.09 |
| Ours | | | 0.20 ± 0.06 | 0.89 ± 0.04 | 0.09 ± 0.03 | **8.80 ± 3.11** | 33.60 ± 1.52 | **0.84 ± 0.05** |
| GES | | | 0.56 ± 0.08 | 0.77 ± 0.07 | 0.40 ± 0.10 | 31.60 ± 7.70 | 52.80 ± 5.72 | 0.54 ± 0.08 |
| PC | 15 | ER2 | 0.62 ± 0.07 | 0.37 ± 0.06 | 0.24 ± 0.03 | 33.40 ± 3.05 | 28.80 ± 1.48 | 0.37 ± 0.07 |
| ICALiNGAM | | | 0.38 ± 0.08 | 0.75 ± 0.08 | 0.18 ± 0.04 | 17.60 ± 3.65 | 36.20 ± 2.49 | 0.68 ± 0.07 |
| NOTEARS | | | 0.17 ± 0.03 | 0.79 ± 0.02 | 0.07 ± 0.01 | 9.80 ± 0.84 | 28.60 ± 0.89 | 0.81 ± 0.02 |
| Ours | | | 0.25 ± 0.04 | 0.99 ± 0.02 | 0.04 ± 0.01 | **6.80 ± 1.48** | 26.60 ± 1.14 | **0.80 ± 0.04** |
| GES | | | 0.43 ± 0.12 | 0.74 ± 0.13 | 0.07 ± 0.02 | 14.90 ± 6.47 | 26.30 ± 1.95 | 0.65 ± 0.13 |
| PC | 20 | ER1 | 0.43 ± 0.08 | 0.61 ± 0.07 | 0.06 ± 0.01 | 13.80 ± 1.92 | 21.60 ± 1.14 | 0.54 ± 0.07 |
| ICALiNGAM | | | 0.47 ± 0.05 | 0.72 ± 0.06 | 0.08 ± 0.01 | 14.60 ± 2.19 | 27.20 ± 1.92 | 0.61 ± 0.05 |
| NOTEARS | | | 0.23 ± 0.08 | 0.79 ± 0.07 | 0.03 ± 0.01 | 8.80 ± 2.86 | 20.60 ± 0.55 | 0.78 ± 0.07 |
| Ours | | | 0.12 ± 0.08 | 0.94 ± 0.02 | 0.04 ± 0.02 | **6.20 ± 3.83** | 43.00 ± 2.83 | **0.91 ± 0.05** |
| GES | | | 0.70 ± 0.07 | 0.64 ± 0.11 | 0.42 ± 0.10 | 65.80 ± 13.57 | 88.80 ± 11.90 | 0.40 ± 0.09 |
| PC | 20 | ER2 | 0.65 ± 0.04 | 0.37 ± 0.06 | 0.18 ± 0.01 | 44.00 ± 1.41 | 41.80 ± 2.39 | 0.36 ± 0.05 |
| ICALINGAM | | | 0.34 ± 0.10 | 0.80 ± 0.06 | 0.11 ± 0.04 | 19.60 ± 6.47 | 49.20 ± 4.21 | 0.72 ± 0.08 |
| NOTEARS | | | 0.15 ± 0.05 | 0.91 ± 0.02 | 0.04 ± 0.02 | 9.40 ± 1.95 | 42.80 ± 2.59 | 0.88 ± 0.03 |
| Ours | | | 0.03 ± 0.09 | 0.99 ± 0.04 | 0.02 ± 0.07 | **0.70 ± 2.21** | 17.40 ± 1.26 | **0.98 ± 0.07** |
| GES | | | 0.39 ± 0.02 | 0.89 ± 0.03 | 0.35 ± 0.05 | 9.80 ± 1.30 | 25.00 ± 1.73 | 0.69 ± 0.02 |
| PC | 10 | SF2 | 0.24 ± 0.03 | 0.69 ± 0.03 | 0.14 ± 0.02 | 7.40 ± 0.55 | 15.60 ± 0.55 | 0.72 ± 0.02 |
| ICALiNGAM | | | 0.40 ± 0.09 | 0.79 ± 0.07 | 0.33 ± 0.09 | 10.80 ± 2.77 | 22.60 ± 1.67 | 0.68 ± 0.08 |
| NOTEARS | | | 0.00 ± 0.00 | 0.82 ± 0.00 | 0.00 ± 0.00 | 3.00 ± 0.00 | 14.00 ± 0.00 | 0.90 ± 0.00 |
| Ours | | | 0.08 ± 0.08 | 0.94 ± 0.06 | 0.17 ± 0.19 | **3.50 ± 3.78** | 30.80 ± 1.87 | **0.93 ± 0.07** |
| GES | | | 0.60 ± 0.05 | 0.56 ± 0.07 | 1.69 ± 0.12 | 27.00 ± 2.35 | 42.20 ± 0.45 | 0.44 ± 0.05 |
| PC | 10 | SF4 | 0.18 ± 0.08 | 0.59 ± 0.06 | 0.25 ± 0.12 | 14.60 ± 2.41 | 21.40 ± 1.52 | 0.68 ± 0.06 |
| ICALiNGAM | | | 0.21 ± 0.06 | 0.83 ± 0.09 | 0.45 ± 0.15 | 9.40 ± 3.21 | 31.80 ± 1.64 | 0.81 ± 0.06 |
| NOTEARS | | | 0.03 ± 0.05 | 0.83 ± 0.09 | 0.05 ± 0.07 | 5.40 ± 3.29 | 25.80 ± 1.64 | 0.89 ± 0.07 |
| Ours | | | 0.02 ± 0.02 | 0.98 ± 0.02 | 0.01 ± 0.01 | **0.60 ± 0.55** | 27.00 ± 0.00 | **0.98 ± 0.02** |
| GES | | | 0.27 ± 0.02 | 0.99 ± 0.02 | 0.13 ± 0.01 | 10.00 ± 1.00 | 36.80 ± 0.84 | 0.79 ± 0.01 |
| PC | 15 | SF2 | 0.15 ± 0.08 | 0.76 ± 0.03 | 0.05 ± 0.03 | 9.40 ± 2.70 | 24.00 ± 1.41 | 0.77 ± 0.04 |
| ICALiNGAM | | | 0.36 ± 0.11 | 0.87 ± 0.07 | 0.17 ± 0.07 | 15.20 ± 7.26 | 37.20 ± 3.83 | 0.74 ± 0.10 |
| NOTEARS | | | 0.02 ± 0.02 | 0.97 ± 0.02 | 0.01 ± 0.01 | 1.40 ± 0.89 | 26.80 ± 0.45 | 0.97 ± 0.02 |
| Ours | | | 0.10 ± 0.06 | 0.94 ± 0.05 | 0.09 ± 0.06 | **7.20 ± 4.55** | 52.20 ± 1.79 | **0.92 ± 0.05** |
| GES | | | 0.47 ± 0.09 | 0.82 ± 0.10 | 0.67 ± 0.16 | 39.40 ± 11.04 | 77.60 ± 4.39 | 0.62 ± 0.09 |
| PC | 15 | SF4 | 0.52 ± 0.06 | 0.29 ± 0.04 | 0.28 ± 0.03 | 45.40 ± 2.51 | 30.00 ± 0.71 | 0.36 ± 0.05 |
| ICALiNGAM | | | 0.44 ± 0.09 | 0.69 ± 0.09 | 0.50 ± 0.11 | 36.40 ± 7.83 | 62.00 ± 3.39 | 0.62 ± 0.09 |
| NOTEARS | | | 0.12 ± 0.03 | 0.83 ± 0.08 | 0.11 ± 0.02 | 13.20 ± 4.92 | 47.40 ± 2.79 | 0.85 ± 0.06 |
| Ours | | | 0.10 ± 0.14 | 0.96 ± 0.05 | 0.03 ± 0.04 | **5.40 ± 7.40** | 40.20 ± 4.38 | **0.93 ± 0.10** |
| GES | | | 0.30 ± 0.12 | 0.96 ± 0.02 | 0.10 ± 0.05 | 16.00 ± 7.28 | 51.20 ± 6.98 | 0.77 ± 0.08 |
| PC | 20 | SF2 | 0.26 ± 0.07 | 0.70 ± 0.06 | 0.06 ± 0.02 | 17.80 ± 4.38 | 34.80 ± 1.48 | 0.68 ± 0.06 |
| ICALiNGAM | | | 0.34 ± 0.17 | 0.87 ± 0.06 | 0.12 ± 0.08 | 19.80 ± 13.01 | 50.80 ± 9.88 | 0.75 ± 0.13 |
| NOTEARS | | | 0.23 ± 0.07 | 0.88 ± 0.04 | 0.06 ± 0.02 | 12.80 ± 5.76 | 42.20 ± 2.28 | 0.82 ± 0.05 |
| Ours | | | 0.06 ± 0.05 | 0.98 ± 0.03 | 0.04 ± 0.03 | **5.60 ± 5.50** | 73.00 ± 1.73 | **0.96 ± 0.04** |
| GES | | | 0.28 ± 0.03 | 0.94 ± 0.04 | 0.21 ± 0.02 | 26.60 ± 3.13 | 91.40 ± 1.52 | 0.79 ± 0.03 |
| PC | 20 | SF4 | 0.38 ± 0.04 | 0.34 ± 0.02 | 0.12 ± 0.01 | 54.60 ± 2.70 | 38.80 ± 1.30 | 0.44 ± 0.03 |
| ICALiNGAM | | | 0.37 ± 0.13 | 0.73 ± 0.05 | 0.27 ± 0.13 | 45.20 ± 17.25 | 83.40 ± 12.46 | 0.67 ± 0.10 |
| NOTEARS | | | 0.18 ± 0.01 | 0.81 ± 0.02 | 0.10 ± 0.01 | 22.80 ± 1.64 | 69.40 ± 0.55 | 0.82 ± 0.02 |

Table 3: Comparison against established structure learning algorithms on various accuracy metrics for ER/SF graphs of increasing complexity. Mean and standard deviation calculated over 10 seeds.

| Dataset | Method | $\gamma$ | $\alpha$ | $\beta$ | $\omega$ | $\eta$ | $p_{ns}$ |
|---------|--------|----------|----------|---------|----------|---------|----------|
| MNIST | DAG | 0.5 | $4e-05$ | $2e-05$ | $8e-4$ | 40.0 | - |
| Fashion-MNIST | DAG | 0.3 | $3e-05$ | $5e-05$ | $1e-3$ | 20.0 | - |
| 2-MNIST | DAG | 0.5 | $4e-05$ | $2e-05$ | $8e-4$ | 40.0 | - |
| MNIST | NS | 0.8 | $1e-04$ | $8e-05$ | 0.05 | - | 0.1 |
| Fashion-MNIST | NS | 0.5 | $1e-04$ | $8e-05$ | 0.05 | - | 0.2 |
| 2-MNIST | NS | 0.8 | $1e-04$ | $8e-05$ | 0.05 | - | 0.1 |

Table 4: Hyperparameters used to obtain the results reported in Fig. 6. *DAG* (directed acyclic graphs) refers to method (1), while *NS* (negative samples) to method (2). The accuracy obtained on 2-MNIST is similar to the one obtained for MNIST. $k$ was set to $1.0$ for negative samples. A weight decay of $0.001$ was applied to the node values during the *NS* experiments.

we observed a wide range of possible output structures (e.g., using zero or multiple hidden-nodes, in parallel or in sequence, with and with-out skip connections) depending on the chosen $\omega$. The best accuracy, however, was always achieved with a structure equivalent to a hierarchical neural network with two fully connected layers as shown in Figure 6. For method (2), instead, the only non-degenerate possibilities were either a complete graph (for low $\omega$ values), the optimal 2-layer network, or a network with no edges (for high $\omega$ values). A possible future research direction could be aiming at combining the two methods to overcome their respective limitations. Table 4 reports the best hyperparameters for each method and dataset.

## H  END-TO-END CAUSAL LEARNING

The goal of the presented experiments so far was to show that a causal predictive coding network can solve both tasks of causality:

- Given observational data, perform unsupervised causal structure learning of the (weighted) adjacency matrix that represents the data generating SCM.
- Given observational data and causal structure, perform inference of associational, interventional, and counterfactual distributions to answer causal queries.

Therefore, this section is motivated by studying the capability of our proposed method to combine both tasks into a single framework. We use the same PC graph to conduct structure learning for the common causal graphs used in the causal inference experiments (*chain, collider, confounder, fork, mediator, butterfly bias, M-bias*) using only observational data generated by the corresponding SCM. Given that our method is able to (i) discover complex causal structures for random DAGs in Section 4 and (ii) correctly answer causal queries for common DAG structures in Section 3, we hypothesize that our method should be able to discover graph structures used in Section 3 without prior knowledge. The motivation behind this approach is that in the real-world, true *causal structures are rarely known* and often only observational data from an *unknown SCM* is available. In the following, we perform causal structure learning for the graphs used in the causal inference experiments of Section 3. The procedure is as follows: First, given observational data, we learn the causal structure of the underlying SCM that generated the observed data. More specifically, we start with a fully connected PC graph and prune unnecessary node connections using sparsity and acyclicity constraints with subsequent thresholding as described in Appendix D. Second, given the observational data and the discovered causal structure, we learn the SCM parameters including an approximation of the parameters of each node's exogenous distribution $\mathbf{u}_i$. To be more specific, once the causal structure is discovered, we modify the PC graph by including one exogenous node, $\mathbf{u}_i$, for each endogenous variable, $\mathbf{x}_i$, into our PC graph. Augmenting the PC graph with exogenous nodes enables us to learn the distribution of each exogenous node, which is crucial in the SCM framework, as exogenous variables are essential for conducting counterfactual inference. This procedure provides us with a simple and closed form end-to-end causal inference engine. Using a single PC model with no pipelines, enables us to (1) discover the adjacency matrix of the SCM and to (2) answer causal queries on any of the three levels of Pearl's ladder of causation (2009). We report results for the structure learning step in Table 5.

| graph | $N$ | MAE $\downarrow$ | FDR $\downarrow$ | TPR $\uparrow$ | FPR $\downarrow$ | SHD $\downarrow$ | NNZ - | F1 $\uparrow$ |
|---|---|---|---|---|---|---|---|---|
| butterfly | 5 | $0.02 \pm 0.01$ | $0.00 \pm 0.00$ | $1.00 \pm 0.00$ | $0.00 \pm 0.00$ | $0.00 \pm 0.00$ | $6.00 \pm 0.00$ | $1.00 \pm 0.00$ |
| M | 5 | $0.03 \pm 0.01$ | $0.04 \pm 0.09$ | $1.00 \pm 0.00$ | $0.03 \pm 0.07$ | $0.20 \pm 0.45$ | $4.20 \pm 0.45$ | $0.98 \pm 0.05$ |
| chain | 3 | $0.01 \pm 0.00$ | $0.00 \pm 0.00$ | $1.00 \pm 0.00$ | $0.00 \pm 0.00$ | $0.00 \pm 0.00$ | $2.00 \pm 0.00$ | $1.00 \pm 0.00$ |
| confounder | 3 | $0.22 \pm 0.12$ | $0.27 \pm 0.15$ | $0.73 \pm 0.15$ | $0.80 \pm 0.45$ | $0.80 \pm 0.45$ | $3.00 \pm 0.00$ | $0.73 \pm 0.15$ |
| collider | 3 | $0.01 \pm 0.00$ | $0.00 \pm 0.00$ | $1.00 \pm 0.00$ | $0.00 \pm 0.00$ | $0.00 \pm 0.00$ | $2.00 \pm 0.00$ | $1.00 \pm 0.00$ |
| fork | 3 | $0.01 \pm 0.01$ | $0.00 \pm 0.00$ | $1.00 \pm 0.00$ | $0.00 \pm 0.00$ | $0.00 \pm 0.00$ | $2.00 \pm 0.00$ | $1.00 \pm 0.00$ |
| mediator | 3 | $0.46 \pm 0.33$ | $0.27 \pm 0.15$ | $0.73 \pm 0.15$ | $0.80 \pm 0.45$ | $0.80 \pm 0.45$ | $3.00 \pm 0.00$ | $0.73 \pm 0.15$ |

Table 5: End-to-end causality engine: Causal predictive coding for discovery of DAGs based on observational data only. Numbers are reported over 5 different runs.

Despite the graphs being very different, the experimental results show that our method performs well in causal discovery for most common causal graphs despite using the same hyperparameters and no hyperparameter search, even though the graphs are very different. We do not show the causal inference results again, because the results for learning associational, interventional, and counterfactual distributions and the distribution of the exogenous noise variables in SCM remained the same. The discovered causal structures are consistent with the adjacency matrices used as prior knowledge in Section 3. Our method is able to solve both causality tasks without prior knowledge of any graph structures. We showed how our causal predictive coding framework can be used in an end-to-end unsupervised causal inference pipeline similar to (Geffner et al., 2022) but without the need of complex neural networks. Thus, our proposed causal predictive coding maintains transparency and interpretability despite good performance.