# OpenReview forum: "Predictive Coding beyond Correlations"
_ICLR.cc/2024/Conference — Submitted to ICLR 2024_

### Official Review · Reviewer_ai27 · 2023-10-30

**Soundness:** 3 good
**Presentation:** 2 fair
**Contribution:** 3 good
**Rating:** 5
**Confidence:** 3

**Summary:**

This paper demonstrates  the potentiality of predictive coding to handle causal related questions including causal inference and causal discovery. In the case of causal inference, the authors set the prediction error term to be 0 and show how this intervention helps the PC based models go beyond correlation. For the structure learning, causal relationships were derived from observational data using PC models. The proposed method is tested on a large number of benchmarks.

**Strengths:**

I think a leap from correlation to causation for predictive coding models could be interesting. This work consider a wide range of causal scenarios that PC models can be useful, including the causal effect estimation, different queries (associational, interventional, and counterfactual), structure learning, and classification. All scenarios are supported by empirical experiments and some are provided theoretical guarantees.

**Weaknesses:**

For the predictive coding go beyond correlation, I was expecting some formal assumptions and theorems to build a solid framework. This paper, however, use more intuitional and descriptive statements to demonstrate the methods. The figures look fancy but not illustrative, I also noticed that Fig. 7 mentioned  in page 5 causal inference section is missing. In addition, the structure is a little bit hard to follow, the important contents are scattered and not cohesive. See more detailed questions below.

**Questions:**

1. page 3: it is noted that the way of removing impact of latent confounder is to do intervention on a randomly selected individual. Then with the intervened data, could we simply do the standard ATE calculation? What is the main advantage of PC model on causal inference compared with other methods?
2. page 4, Theorem 3.1: Any constraints on the horizon t? How is the asymptotic performance of the proposed estimator? Any sample size calculation for the PC related method?
3. page 6, classification: Could you further explained the relationship between classification and causal inference under the predictive coding model setting? What are the classes here? And how test accuracy is related to causal effect?
4. If the graph is unknown, could the PC model still do causal inference by using the structure learned by PC from observational data?

---

> ### Author Response · Authors · 2023-11-20
>
> We thank the reviewer for its time and effort.
>
> >  What is the main advantage of PC model on causal inference compared with other methods?
>
> While our method sometimes performs better than existing ones, the main goal of our work is not to solve open problems in the causal inference literature, but to provide a bridge between predictive coding (an influential theory in neuroscience), and the field of causal inference. Hence, we foresee this work to have an impact in the predictive coding community, as it extends the toolbox of computational neuroscientists to contain techniques that allow them to perform causal inference and structure learning, which was never the case before: So far if people wanted to use predictive coding models to perform causal inference, they would have to train a new model for every intervention variable (because it is a different graph after mutilation/intervention). however, we show how the original graph can be modified in the causal inference step in order to render equivalent causal conclusion without the need to train a new model for every intervention variable
>
> > Could you further explained the relationship between classification and causal inference under the predictive coding model setting? What are the classes here? And how test accuracy is related to causal effect?
>
> The classes are the labels of the MNIST dataset.  The reason why interventions improve the test accuracy on classification tasks is that they prevent the error of the inputs to spread in the network, and hence enforce a specific direction of the information flow, which goes from cause (the image) to effect (the label).
>
> > If the graph is unknown, could the PC model still do causal inference by using the structure learned by PC from observational data?
>
> Yes and this can be done without changing the model parameters or the architecture. Thus one could see it as an end to end causal inference engine as shown in our experiments of Appendix H. All that is required is to set the error node of the intervention variable to zero and then perform inference.

---

### Official Review · Reviewer_sN6J · 2023-10-31

**Soundness:** 3 good
**Presentation:** 2 fair
**Contribution:** 2 fair
**Rating:** 6
**Confidence:** 3

**Summary:**

This paper introduces a method for causal inference using predictive coding. The authors demonstrate that, given a known causal graph, the proposed method can perform both interventional and counterfactual inferences. In cases where the causal graph is unknown, a gradient-based approach is introduced to discern the causal structure. Through experiments with synthetic data, the authors illustrate that their method outperforms existing approaches.

**Strengths:**

- The paper is self-contained with a pretty novel approach for causal inference.
- The proposed method outperforms existing methods for structural learning, interventional, and counterfactual inferencing.
- The authors claim that the proposed method is parameter efficient and does not require extensive hyperparameter tuning.

**Weaknesses:**

- It is not very obvious how the proposed method compares with the existing frameworks. So, it is also not very clear how the proposed method would contribute to the research direction. An in-depth discussion comparing the PC graph with DAG might lend more weight to the paper's influence on the causality.
- It is not easy to see in principle how the proposed method is superior to the probabilistic method.
- While the comparison against baseline methods using synthetic data provides some insights, it would be more convincing to see the comparison of inferencing and structural learning with some real data.

**Questions:**

- In equation 1, is the variable $x$ sampled from the standard normal distribution or a parameter of the model?
- Is there any intuition of how the PC graph outperforms the existing method? Is it mainly because the proposed model better estimates the observational probabilities?

---

> ### Author Response · Authors · 2023-11-20
>
> We thank the reviewer for its time and feedback on the manuscript.
>
> > How can it contribute to the research direction?
>
> The main goal of our work is not to solve open problems in the causal inference literature, but to provide a bridge between predictive coding (an influential theory in neuroscience), and the field of causal inference. Hence, we foresee this work to have an impact in the predictive coding community, as it extends the toolbox of computational neuroscientists to contain techniques that allow them to perform causal inference and structure learning, which was never the case before.
>
> The assumptions that we make are coherent with this line of argumentation. For example, we assume causal sufficiency, and we only compare against the large amount of works in causal inference that make this assumption. This is quite common in a branch of the causal inference literature (e.g., Geffner Thomas, et al. "Deep end-to-end causal inference, 2022).
>
> > In equation 1, is the variable $x$ sampled from the standard normal distribution or a parameter of the model?
>
> It is a latent variable of the model, optimized via gradient descent.  We initialize it via sampling. However, the best way this parameter can be initialized is currently an open question.
>
> > Is there any intuition of how the PC graph outperforms the existing method? Is it mainly because the proposed model better estimates the observational probabilities?
>
> Yes: We believe that the flexibility (and generality) of our method made by hierarchies of Gaussians, but optimized locally, allows to better estimate the observational probabilities.

---

### Official Review · Reviewer_9K6b · 2023-11-01

**Soundness:** 3 good
**Presentation:** 2 fair
**Contribution:** 2 fair
**Rating:** 5
**Confidence:** 4

**Summary:**

The authors show that it is possible to use predictive coding on graphs to answer causal and counterfactual queries as well as learn the causal structure. For causal queries, this is done by mimicking interventions (fixing certain nodes to a value) during the inference phase of PC. For counterfactual queries the process is similar - the usual operations of abductions, action and prediction are shown to be do-able with PC inference. Structure learning can also be done using PC on the adjacency matrix of the graph.

**Strengths:**

- The paper provides another way of answering causal queries from the conventional methods, which is interesting in itself

**Weaknesses:**

- The main weakness of the work is the presentation, in that it is very hard to parse the information. A lot of the details are in the Appendix but there are not enough pointers to the Appendix in the main text (e.g. Appendix D is never referenced).
	- Where are the results comparing the causal queries with VACA, CAREFL, etc? It's not stated in the main text.
	- Results in section 3.2. For which experiment are these results of? It seems its only for counterfactual queries.
	- Figures like fig 4, fig 5, and fig 6 have too much information from too many results. These results can also be presented in much cleaner formats. For example, in structure learning the mean squared error of the learned and actual adjacency averaged over multiple runs would be much more informative than simply showing the adjacency for two graphs.
	- Figures 15- 20: Its unclear for what tasks and under which data generation regime these results are for.
	- Figures 8 -14: I can't see these referenced anywhere and so it is entirely unclear what these are showing.


- Another main weakness is that while it is interesting that it is possible to perform causal queries with PC, what is the obvious advantage? For causal queries, contradictory to the claims made in the paper, its not obvious that the PC approach is more efficient (see below). For structure learning, why would PC perform better than other maximum likelihood methods? A more thorough discussion of what is exciting about the fact that PC can be used to answer causal queries would benefit the reader.

**Questions:**

- It should be made more clear that the method assumes that the observed variables are Gaussian distributed (eq 1).
- Intervention query: Am I correct in assuming that for each intervention query, a new PC graph has to be trained from scratch? I believe this is the case as a different error has to be zeroed out. If so, is this not very inefficient? This is not the case if interventional queries are found by doing graph mutilations.
- Classification task: A lot more detail is needed for this to be clear. It is simply stated that "we perform an intervention on the input" but its not clear what this means. An intervention in a causal sense means, putting in another input which will change the label as well. If you mean zeroing out the errors of the input nodes, this needs to be made clear. "Intervention" has a very specific meaning in causal literature. The Appendix has insufficient details.
- Structure learning: Has the data been normalised? If not, this should be mentioned and compared against standardised data in line with [1]

Minor points:
- X_unk is never formally defined
- Appendix is a bit messy.
	- Took a while to find where the non-linear data generation was as Appendix D is never referred to in any of the text.
	- Final line in discussion in page 20: This seems slightly misleading, the data in all cases is generated with additive Gaussian noise, whereas the methods discussed are specifically designed to deal with Non-Gaussian distributions.

- How does predictive coding handle a single dataset? All the theory in the paper assumes access to a single datapoint.
- Section 3, Interventional query: "...perform an intervention", what variable the intervention on? It's not clear here
- Section 2.1: What is time step t refering to? It has not been defined.
- Given that each x_i is a node in the (causal) graph, how is it possible to use deeper neural networks (eq 1) to learn the relationship between observed variables? The current formulation seems to limit the relationship between variables to 1 layer NNs.
- If generated from non linear data, the counterfactual queries are not identifiable, how does the method handle this?

[1] Reisach, Alexander, Christof Seiler, and Sebastian Weichwald. "Beware of the simulated dag! causal discovery benchmarks may be easy to game." Advances in Neural Information Processing Systems

---

> ### Author Response · Authors · 2023-11-20
>
> We thank the reviewer for its time and comments.
>
> > It should be made more clear that the method assumes that the observed variables are Gaussian distributed (eq 1).
>
>  Note that this is not the case: It is an assumption of the variables of the model, but it doesn’t need to be true for the input data, otherwise we wouldn’t be able to train a classification network for which the output is a one-hot encoded vector. This is true only for the variables of the model itself, that is an hierarchy of Gaussian.
>
> > Am I correct in assuming that for each intervention query, a new PC graph has to be trained from scratch?
>
> This is also not true: once the model is trained on a specific dataset,, it is possible to perform all the interventions needed, as they do not interfere with the parameters learned.
>
> > Classification task: A lot more detail is needed for this to be clear. It is simply stated that "we perform an intervention on the input" but its not clear what this means.
>
> We will state this more clearly in a future draft of the manuscript. However, performing an intervention on the input simply means fixing the input nodes to the input values, and zeroing out their errors.
>
> > Has the data been normalised?
>
> The data has not been normalized. We will add this explanation.
>
> > How does predictive coding handle a single dataset?
>
> As every other deep learning method: training can be achieved via SGD applied on mini (or full)-batches to minimise the average energy F.
>
> > Section 2.1: What is time step t refering to? It has not been defined.
>
> We will update the description to clarify this. It refers to a single iteration of gradient descent on the value nodes x.
>
> > Given that each x_i is a node in the (causal) graph, how is it possible to use deeper neural networks (eq 1) to learn the relationship between observed variables?
>
> Using hierarchical generative models. This is how we have performed the experiments on the colored MNIST dataset.
>
> > In structure learning the mean squared error of the learned and actual adjacency averaged over multiple runs would be much more informative than simply showing the adjacency for two graphs.
>
> This is merely for a space reason: we have decided to report single metrics in the supplementary material, as every table is as large as a single page.
>
> > pointers on clarity
>
> Thank you for all the comments and pointers about the presentation of our work: We will address them all, and they will largely improve the manuscript. We had removed most of the pointers from the main body of the manuscript to the supplementary material, to not refer to the supplementary material in every paragraph. However, at the beginning of the supplementary material, there is a detailed index, divided in the same way as the sections of the main paper, that states where to find the experiments of every topic.

---

> > ### Comment · Reviewer_9K6b · 2023-11-22
> > **Response**
> >
> > Thanks for you response and clearing up some my misunderstandings.
> >
> > Re efficiency: I was referring to the line "To avoid this and perform an intervention, we set the value of e2 to zero
> > throughout the inference phase". Am I mistaken in thinking that this implies that the inference has to be rerun for a different error to be set to zero (and hence a different query)? I'm not sure what parameters you are referring to in your response, I assumed the parameter of the dirac delta variational approximation have to be retrained.
> >
> > Re Gaussianity: As in a regular causal graph, I assumed that each vertex corresponds to an observed variable. Hence making the assumption that each conditional is Gaussian distributed is restricting what you can model. I do agree the variables themselves do not have to be Gaussian.
> >
> > Re normalisation: The paper I referenced shows that methods like No-Tears can fail when data is normalised as its performance is due to exploiting the variance of data that is a common feature of simulated experiments (but will not help in real world cases). It's hard to consider results without discussion of this artefact.
> >
> > My concerns still remain - mainly in the presentation of the material which hinders its understanding, and the contribution as its still not clear why we should be excited that it is possible to perform causal tasks with predictive coding. The presentation can be made a lot clearer, and a strong argument for why predictive coding can help causality (for example clearly explaining the efficiency) would help.

---

### Official Review · Reviewer_ER1E · 2023-11-01

**Soundness:** 1 poor
**Presentation:** 1 poor
**Contribution:** 2 fair
**Rating:** 3
**Confidence:** 3

**Summary:**

The paper proposes the introduction of the predictive coding for causal inference and discovery. They claim that a few simple changes in the standard PC process enable these both these tasks.

**Strengths:**

- Original exploration of PC concepts in this context

**Weaknesses:**

- Paper is very informally written and theorems / definitions are not up to standards of a statistics community. Eg Theorem 1 has no assumptions stated, what are the spaces allowed of the variables involved etc.
- **I don't see actual methodological novelty wrt to causal inference and/or discovery except "interpretations" of existing techniques in the parlance of predictive coding.** If the interpretation is the only contribution, can you clarify what is the value here?
- Theorem 1 is irrelevant and not new or a contribution, I am not sure why it is there.  The claim right before it "This approach obviates the need for explicit adjustment formulas and back-door criteria in
causal inference." is misleading and hints at a contribution where there isn't one. If I understand correctly, the claim is true simply as an immediate consequence of the "truncated factorization" (Pearl 2009, Eq 1.37) when no hidden confounders are involved. In this case, there is no need for back-door adjustments or other similar things already by standard causal inference.
- Same for Sec 2. Causal graph is defined wrongly or incomplete at least; the Markov factorization is not what makes an CBN, since it applies to standard BNs. See definitions of CBN in eg [Pearl, 2009, Definition 1.3.1] , it's about all interventional distributions having the Markov fact. and other constraints.
- What are the assumptions in the part before 2.1  "Posterior distribution" ? The method is presented in the intro/abstract as very general but here we seem to require Gaussian assumptions, unit variances, mean field approximations .. all these in text and not presented properly mathematically (see comment above too). Is the setting the same as Peters et al (see below reference) ?
- At start of 2.1 you say you will use SGD for optimizing ( I guess F from Eq 2) - why SGD ? This is a simple least squares objective to optimize and simpler (faster) alternatives exist. Unless I'm missing something but again not very formally presented, there is no clear optimization problem stated with argmins etc.
- For discovery you use the method of Zhang et al (2018) - again where is the novelty here ? Is PC used to propose a different method or not ?


References
- Peters, J. and Bühlmann, P., 2014. Identifiability of Gaussian structural equation models with equal error variances. Biometrika, 101(1), pp.219-228.

**Questions:**

Explained in the weaknesses section

---

> ### Author Response · Authors · 2023-11-20
>
> We thank the reviewer for the comments and the detailed feedback.
>
> >Paper  is very informally written and theorems / definitions are not up to standards of a statistics community.
>
> The assumptions are stated in section 2, and referred before the theorem, i.e. the distributions we are referring to are the gaussians approximated with dirac deltas. We did not re-state them at the beginning of the theorem due to a lack of space, and avoid repetitions.
>
>
>
> > Theorem 1 is irrelevant and not new or a contribution:
>
> The reason we believe the theorem is relevant is because it formally claims that performing causal inference on a mutilated predictive coding graph is equivalent to performing regular inference on the original predictive coding graph after setting the error node of the intervention variable to zero.
>
> >  The claim right before it "This approach obviates the need for explicit adjustment formulas and back-door criteria in causal inference." is misleading
>
> Our intention with the statement was in reference to the exchangeability assumption and how it allows us to identify a causal effect. Exchangeability assumes that the graph is in a mutilated form (which is not the case for predictive coding networks because we never perform such a graph mutilation, and keep the original predictive coding graph). Therefore, in classical causal inference one has to adjust for the observed confounder in order to achieve conditional exchangeability. However, in our case it suffices to set the error node of the intervention variable to zero in order to render the mutilated graph in which exchangeability holds again. To ensure clarity and accuracy, we will revise our statement to reflect that our method’s utility is in its application to scenarios with observed confounders
>
>
>
> > What are the assumptions in the part before 2.1 "Posterior distribution" ?
>
> Section 2 is a summary of the framework of predictive coding as defined in the referenced works. Consequently, we make use of the same assumptions to arrive at Eq. 2 and algorithm 1. Going through the details of derivations would take us the whole manuscript, and hence we refer to the many works that have defined it.  Regarding the assumptions you mentioned, they are standard in predictive coding but not limiting (for a reference, see for example [5], [6]). Here, we show that the standard framework is sufficient to perform in the proposed benchmarks. In particular, the assumptions listed in this section, refer only to the underlying probabilistic assumptions used to describe predictive coding under the lens of variational inference and derive eq. 2.
> Regarding the question if our setting is the same as Peters et al. [2018], the answer is no as their work focuses solely on learning the underlying directed acyclic graph with linear functions. Our setting is different as we experiment with both linear and nonlinear functions as well as structure learning and causal inference. Furthermore, we don’t claim full identifiability, instead our method of structure learning identifies the graph from the joint distribution only up to Markov equivalence class assuming faithfulness instead of weak faithfulness (causal minimality).
>
> > At start of 2.1 you say you will use SGD for optimizing ( I guess F from Eq 2) - why SGD ?
>
> Eq. 2 is a complex non-convex function and its optimization is currently an open and quite active area of research (see [3], [2]). Currently stochastic gradient descent is the go-to method to optimise such models.  In terms of the optimization problem, our goal is to learn the set of weights W that allow us to compute the posterior p(X_unk|X_data = S_data) for any given S_data. This is achieved, as explained in section 2, by minimising the energy F defined in eq. 2 with respect to the node values X and the weights W as described in algorithm 1. Formally, during training, we are looking for W*, defined as:
>
> X* = argmin_{X}F(X,W)
> W* = argmin_{W}F(X*,W)
>
> Where X is the training data and W the set of weights defining the PCN. This is standard in PC literature (see for example [1], section 2.1) and the optimization problem is commonly tackled with a combination of Expectation-Maximization and SGD.

---

> > ### Author Response · Authors · 2023-11-20
> >
> > > For discovery you use the method of Zhang et al (2018) - again where is the novelty here ? Is PC used to propose a different method or not ?
> >
> >  We show that the method can be used in combination of the free energy method of predictive coding. By doing so, we can learn both the adjacency matrix (i.e., which node is connected to which node) using the method proposed in Zhang et al, but also which complex relation links two nodes (given that an edge in a pc graph can represent non-linear functions), differently to Zhang et al, where the adjacency matrix only represents a set of coefficients by which each node is multiplied.
> > We do not claim this to be novel, however, a key finding of our experiments is the superior performance of our methodology over NOTEARS in various experimental settings, particularly in Scale Free graphs. We attribute this enhanced performance to the inherent properties of predictive coding networks. Further, our decision to integrate structure learning was driven by a desire to demonstrate to the neuroscience communities the adaptability of predictive coding networks. Their use facilitates end-to-end causal inference approaches, enabling a consistent methodology without the need for significant alterations.  We hope this clarification underscores the significance of our approach in contributing to the broader understanding of structure learning in potentially complex networks.
> >
> > [1] Millidge, Beren, Anil Seth, and Christopher L. Buckley. "Predictive coding: a theoretical and experimental review." arXiv preprint arXiv:2107.12979 (2021).
> >
> > [2] Alonso, Nick, Jeff Krichmar, and Emre Neftci. "Understanding and Improving Optimization in Predictive Coding Networks." arXiv preprint arXiv:2305.13562 (2023).
> >
> > [3] Salvatori, Tommaso, et al. "Predictive coding can do exact backpropagation on convolutional and recurrent neural networks." arXiv preprint arXiv:2103.03725 (2021).
> >
> > [4] Salvatori, Tommaso, et al. "Learning on arbitrary graph topologies via predictive coding." Advances in neural information processing systems 35 (2022): 38232-38244.
> >
> > [5] Whittington, James CR, and Rafal Bogacz. "An approximation of the error backpropagation algorithm in a predictive coding network with local hebbian synaptic plasticity." Neural computation 29.5 (2017): 1229-1262.
> >
> > [6] Pinchetti, Luca, et al. "Predictive Coding beyond Gaussian Distributions." arXiv preprint arXiv:2211.03481 (2022).

---

> ### Author Response · Authors · 2023-11-20
>
> > I don't see actual methodological novelty wrt to causal inference and/or discovery
>
> The main goal of our work is not to solve open problems in causal inference and/or discovery, but to provide a bridge between predictive coding (an influential theory in neuroscience), and the field of causal inference. Hence, we foresee this work to have an impact in the predictive coding community, as it extends the toolbox of computational neuroscientists to contain techniques that allow them to perform causal inference and structure learning, which was never the case before. Hence, we would like our work to be evaluated in terms of how impactful our findings are for the computational neuroscientist, and for the machine learning researcher that is exploring training algorithms that are alternative to backpropagation, rather than how well our method is suited to solve open problems in the causal inference literature (which is beyond the scope of this work). This is why we have selected
>  applications to neuroscience & cognitive science as a primary area of our work.

---

### Meta-Review · Area_Chair_G75h · 2023-12-06

**Metareview:**

The paper proposes a modification to predictive coding that facilitates interventional and counterfactual inference under a known (or unknown) causal graph.

pros:
+ The paper provides a unique perspective bridging predictive coding and causal inference.

cons:
+ lack clear and sufficient comparisons with existing causal inference frameworks
+ lack of formal characterization and sufficient details in definitions and assumptions
+ lack of sufficient real data empirical studies

**Justification For Why Not Higher Score:**

lack of formal theoretical foundation that is essential for claiming causal inference; lack of sufficient empirical studies

**Justification For Why Not Lower Score:**

NA

---

### Decision · Program_Chairs · 2024-01-16

Reject